# RIF1 regulates early replication timing in murine B cells

Daniel Malzl[1,2,8], Mihaela Peycheva ●[1,6,8], Ali Rahjouei[3], Stefano Gnan ●[4], Kyle N. Klein[5], Mariia Nazarova ●[1], Ursula E. Schoeberl ●[1], David M. Gilbert ●[5], Sara C. B. Buonomo ●[4], Michela Di Virgilio ●[3], Tobias Neumann ●[1,7] ✉ & Rushad Pavri ●[1] ✉

The mammalian DNA replication timing (RT) program is crucial for the proper functioning and integrity of the genome. The best-known mechanism for controlling RT is the suppression of late origins of replication in heterochromatin by RIF1. Here, we report that in antigen-activated, hypermutating murine B lymphocytes, RIF1 binds predominantly to early-replicating active chromatin and promotes early replication, but plays a minor role in regulating replication origin activity, gene expression and genome organization in B cells. Furthermore, we find that RIF1 functions in a complementary and non-epistatic manner with minichromosome maintenance (MCM) proteins to establish early RT signatures genome-wide and, specifically, to ensure the early replication of highly transcribed genes. These findings reveal additional layers of regulation within the B cell RT program, driven by the coordinated activity of RIF1 and MCM proteins.

The faithful and timely replication of the genome is essential for inheriting genetic information and avoiding chromosomal abnormalities. To ensure this, large metazoan genomes initiate replication from several discrete loci, termed origins of replication. Origins are not activated simultaneously across the genome, but rather, in an asynchronous manner referred to as the DNA replication timing (RT) program[1]. A hallmark of the RT program is that genomic A compartments enriched in transcriptionally active genes generally replicate earlier in S phase whereas B compartments harboring silent heterochromatin typically replicate later in S phase[2,3]. Deregulation of RT has been correlated with defects in chromosome condensation, sister chromatid cohesion, gene expression and genome instability[4–6]. Altered RT can disrupt the distribution of active and repressive epigenetic marks causing major alterations in genome architecture[7]. In genetic diseases and cancer, defects in RT have been correlated with deleterious chromosomal translocations[8–12]. Recently, RT has been directly implicated in the

biogenesis of oncogenic translocations found in B cell lymphomas and other leukemias[13]. In addition, late replication has been consistently associated with higher rates of mutation across species[11,14–17] and stress-induced delays in replication are a hallmark of common fragile sites in long, transcribed genes[18–20]. Yet, despite its key role in maintaining cellular physiology and genome integrity, our understanding of the mechanisms regulating RT remains incomplete.

Origins are specified in the late G2/M and early G1 phases of the cell cycle via the loading of the origin recognition complex (ORC) and associated factors[21–25]. The subsequent recruitment of the hexameric ring-shaped MCM complex helicase (MCM2-7) licenses these sites for activation[22], resulting in the formation of the pre-replication complex (pre-RC)[21–25]. In S phase, a subset of licensed origins are activated by a set of proteins collectively called replication firing factors, resulting in the initiation of replication[21–24,26]. This has led to the idea that firing factors must be recycled to activate subsequent sets of origins later in S phase,

[1]Research Institute of Molecular Pathology (IMP), Vienna Biocenter, 1030 Vienna, Austria. [2]CeMM Research Center for Molecular Medicine of the Austrian Academy of Sciences, 1090 Lazarettgasse 14, Vienna, Austria. [3]Max-Delbruck Center for Molecular Medicine in the Helmholtz Association (MDC), 13125 Berlin, Germany. [4]School of Biological Sciences, Institute of Cell Biology, University of Edinburgh, Edinburgh EH9 3FF, UK. [5]San Diego Biomedical Research Institute, San Diego, CA 92121, USA. [6]Present address: CeMM Research Center for Molecular Medicine of the Austrian Academy of Sciences, 1090 Lazarettgasse 14, Vienna, Austria. [7]Present address: Quantro Therapeutics, Vienna Biocenter, 1030 Vienna, Austria. [8]These authors contributed equally: Daniel Malzl, Mihaela Peycheva. ✉e-mail: tobias.neumann@imp.ac.at; rushad.pavri@imp.ac.at

thus ensuring timely completion of replication. In support, over-expression of firing factors has been shown to advance the RT of late-replicating chromatin[27–29]. These studies imply that the time of recruitment of the firing factors to a licensed origin determines its RT[30]. In essence, the probability of origin activation in S phase determines the RT of a region, meaning that early RT domains harbor origins that typically fire earlier in S phase whereas late RT domains contain origins that tend to fire later in S phase.

The best-studied mechanism of RT control is the suppression of late origin firing by the multifunctional protein, RIF1[31–33]. RIF1 associates with late-replicating chromatin in large, mega-base domains called RIF1-associated domains (RADs) and represses origin firing via protein phosphatase 1 (PP1)-mediated dephosphorylation of MCM4[34–36]. Ablation of RIF1 (Rif1−/−) in human and mouse embryonic stem cells (hESCs and mESCs) and in primary mouse embryonic fibroblasts[31] resulted in a genome-wide loss of early and late RT domain distinction associated with alterations in epigenetic marks and genome compartmentalization[7,35,36]. However, in some Rif1−/− cell lines, this RT phenotype was considerably weaker[7], suggesting the existence of additional modes of RT control.

We recently showed that in MCM-depleted CH12 cells, a murine B cell line, the RT program was globally deregulated without major changes in transcription or genome architecture[13]. Since RIF1 is the only other factor whose loss could lead to such a major phenotype[7,31,35], we investigated the role of RIF1 in the B cell RT program. We report the surprising findings that RIF1 is predominantly bound to active chromatin in B cells and promotes their early replication. In addition, we find that RIF1 acts in a complementary and non-epistatic manner with MCM complexes to drive early replication, especially of highly transcribed regions. In sum, our study reveals an additional regulatory layer within the global RT program and a role for RIF1 in promoting early replication.

## Results

### RIF1 regulates early replication in B cells

To measure RT, we performed Repli-seq from early (E) and late (L) S phase fractions in normal and Rif1−/− CH12 B cells[37,38] (Supplementary Fig. 1a). MCM complexes were depleted by infection with lentiviruses expressing short hairpin RNAs (shRNAs) targeting Mcm6 (shMcm6), with an shRNA against LacZ (β-Galactosidase) (shLacZ) serving as a control for infection and shRNA expression, as described previously[13,39] (Supplementary Fig. 1b). RT was calculated as the (log2) E/L ratio[40] (Fig. 1a). RT values showed the expected bimodal distribution in shLacZ cells reflecting distinct early (E; positive RT) and late (L; negative RT) domains (Fig. 1a), which was also reflected in the genomic RT profiles (Fig. 1b). In shMcm6 cells, this distinction was globally weakened with all early and late RT values approaching zero (Fig. 1a, b). In Rif1−/− shLacZ cells, many early-replicating domains underwent a delay in replication (Fig. 1b). In comparison, late RT domains showed a mixed phenotype in Rif1−/− shLacZ cells with some, typically smaller, domains showing advanced replication signatures, and other, typically larger, domains undergoing delayed replication relative to shLacZ cells (Fig. 1a, b).

To directly compare RT between the different conditions, we divided the genome into 20 kb bins and generated RT scatter plots. The results showed that most early- and late-replicating bins were strongly shifted towards zero in shMcm6 cells, indicative of a deregulation of the RT program (Fig. 1c). In Rif1−/− shLacZ cells, most early-replicating regions underwent delayed replication to varying degrees (note the red stripe in Fig. 1c), albeit to a lesser extent than in shMcm6 cells. By contrast, late-replicating bins were much less affected in Rif1−/− shLacZ cells although some bins with slightly delayed RT and others with slightly advanced RT were visible (Fig. 1c).

We also performed Repli-seq in a more physiological system, namely, primary activated splenic B cells from Rif1+/+, RIF1 heterozygous (Rif1−/+) and Rif1−/− mice[41] (Supplementary Fig. 1c). WT primary B cells had relatively fewer early replicating domains and more mid-replicating

domains than shLacZ CH12 cells, resulting in different RT density profiles (Fig. 1d, compare with Fig. 1a). Strikingly, we observed a nearly exclusive effect on early-replicating genomic bins in heterozygous Rif1−/+ cells indicating that reduced levels of RIF1 protein was sufficient to delay replication of early-replicating regions without majorly affecting late replicating ones (Fig. 1d, e). This phenotype was exacerbated in homozygous Rif1−/− cells as seen by the further delay in replication of early RT domains, which was accompanied by the earlier replication of several late RT domains albeit to varying degrees (Fig. 1b, d, e). These results, especially from heterozygous cells, highlight a role for RIF1 in positively regulating early replication in B cells.

In other cell lines, RIF1-PP1 was shown to delay late origin firing by dephosphorylating MCM4[34–36]. Moreover, the RIF1-PP1 interaction has been reported in B cells[38]. Therefore, we asked whether the positive regulation of origin firing in B cells was due to defective MCM4 targeting by RIF1-PP1. In previous work with Rif1−/− or RIF1-knockdown human cells, a portion of MCM4 in the chromatin fraction was found to exist in a hyperphosphorylated state, manifesting as a slower migrating species in SDS-PAGE assays[42,43]. In agreement, we observed that a small fraction of chromatin-associated MCM4 in Rif1−/− CH12 cells was migrating slower than the major MCM4 band (Supplementary Fig. 1d, lanes 3-4). In contrast, MCM4 in WT cells manifested as a single faster-migrating band (Supplementary Fig. 1d, lanes 1-2). Importantly, there was no discernible difference in the levels of chromatin-associated MCM4 between WT and Rif1−/− cells (Supplementary Fig. 1d, compare lanes 1-2 with 3-4). Upon addition of tautomycin, which inhibits PP1 activity, a slower-migrating fraction was also observed in WT cells (Supplementary Fig. 1d, lanes 5-6), implying that the slower-migrating fraction of MCM4 in Rif1−/− cells corresponds to hyperphosphorylated MCM4 resulting from the lack of the RIF1-PP1 interaction. These observations suggest that the absence of the late replication phenotype in B cells is not associated with deficient MCM4 dephosphorylation via RIF1-PP1.

To systematically measure changes in RT upon loss of RIF1, we generated three RT states, early (E), middle (M) and late (L), in shLacZ CH12 cells using a Hidden Markov Model (HMM) to segment the genome into 20 kb bins based on their RT value (Fig. 1f). We split the M state at the zero RT value into early-like (E-like) having positive RT values and late-like (L-like) having negative RT values, which resulted in four RT classes (Fig. 1f). To allow direct comparison of RT changes between conditions, we created density plots for each of the four RT states in shLacZ cells displaying the RT of all other experimental conditions in those genomic bins. The results showed that E and E-like bins in shLacZ replicated later in Rif1−/− cells, but, importantly, that this was not accompanied by commensurate advances in the RT of L bins suggesting that loss of RIF1 does not globally deregulate RT (Fig. 1g). Indeed, most L bins retained their RT values with only some bins showing advanced RT and some others showing further delays in RT relative to shLacZ cells (Fig. 1g). In contrast, in shMcm6 cells, all shLacZ RT bins shifted towards zero RT values implying that these cells have undergone a global deregulation of the RT program (Fig. 1g).

The same HMM-based analysis in primary, splenic B cells revealed that in Rif1−/+ heterozygotes, there was a strong delay in the RT of E bins associated with a much weaker advance in the RT of L bins and virtually no changes in L-like bins (Fig. 1h, i). These changes were exacerbated in Rif1−/− B cells as seen by the further delay in the RT of E bins accompanied by the advanced RT of L bins relative to Rif1−/+ cells (Fig. 1h, i). The profile of RIF1 heterozygous cells suggests that the major role of RIF1 in the B cell RT program is to promote early replication of active chromatin domains. It is plausible that the advanced RT of late RT domains in Rif1−/− cells arise, in part, due to indirect effects stemming from having fewer early-firing origins that could lead to the advanced firing of normally late origins.

Altogether, we conclude that RIF1 functions as a modulator of early RT in B cells.

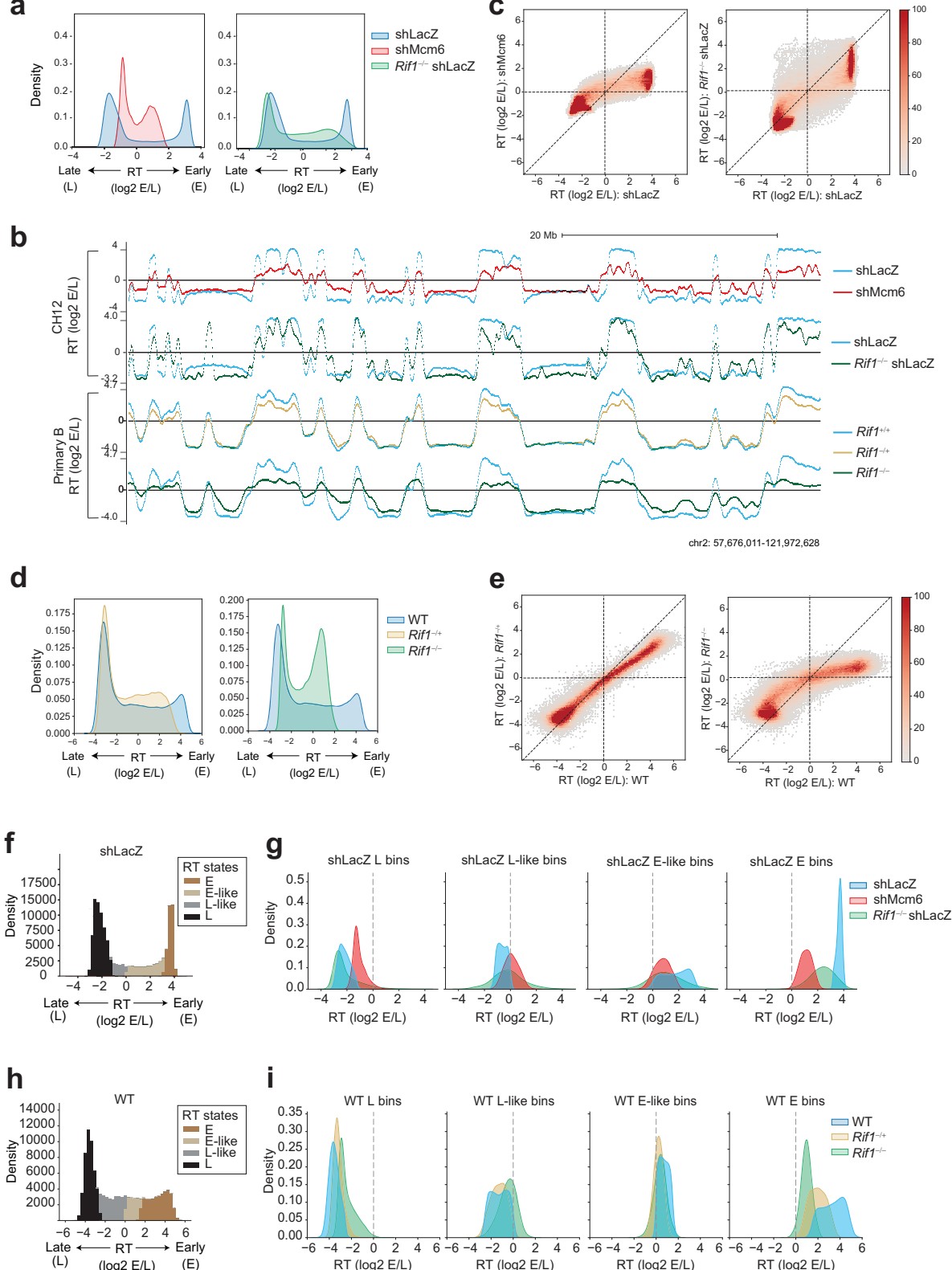

## RIF1 plays a minor role in gene expression and genome compartmentalization in B cells

To determine the role of RIF1 in gene expression, we performed RNA-seq from two independent clones (clones 1 and 2) of *Rif1*−/− CH12 cells and identified differentially regulated genes with the DE-seq2 software[44] (Supplementary Fig. 2a, b). Only 22 downregulated and 3 upregulated genes were common between the two clones (Supplementary Data 1). A gene ontology (GO) analysis of the downregulated

genes did not reveal enrichments in pathways related to DNA replication, cell division, cell proliferation or genome organization (Supplementary Fig. 2c), and there were no pathways enriched in the upregulated genes.

RIF1 has been implicated in regulating genome compartmentalization, in part, through the deregulation of RT[7]. However, in *Rif1*−/− shLacZ CH12 cells, Hi-C analysis revealed no gross changes in compartment profiles based on evaluation of Hi-C heatmaps where

**Fig. 1 | RIF1 regulates early RT in activated B cells. a** RT histograms in 5 kb genomic bins from shLacZ, shMcm6 and *Rif1*−/− shLacZ CH12 cells. RT is calculated as the ratio of the read densities of early (E) and late (L) Repli-seq fractions (log2 E/L). **b** UCSC genome browser view of RT for the indicated conditions in CH12 cells (top panel) and primary, activated splenic B cells (bottom panel). Positive and negative values correspond to early and late-replicating regions. **c** Comparison of RT values in 20 kb genomic bins between shLacZ and shMcm6 CH12 cells (left) and shLacZ and *Rif1*−/− shLacZ cells (right). **d** RT histograms as in **a** from WT, *Rif1*−/+ and *Rif1*−/− primary, activated splenic B cells. **e** Comparison of RT values in 20 kb genomic bins between WT, *Rif1*−/+ and *Rif1*−/− in primary B cells. **f** Classification of shLacZ RT bins from CH12 cells into four states using a Hidden Markov Model (HMM). Three states E, M (mid) and L were called, and the middle state was split into into E-like and L-like at the log2 (E/L) = 0 boundary. **g** RT histograms showing how the bins in each shLacZ RT state (blue) called in **d** change in shMcm6 (red) and *Rif1*−/− shLacZ (green) CH12 cells. **h** HMM analysis as in **d** but using WT RT values from primary B cells to call RT states. **i** RT histograms showing how bins in each WT RT state in primary B cells (blue) change in *Rif1*−/+ (ochre) and *Rif1*−/− (green) cells.

compartments manifest as the checkerboard patterns off the diagonal (Fig. 2a). We quantified the compartment signals based on the first principal component (PC1) of the Hi-C contact matrix where positive PC1 values denote active, early-replicating A compartments and negative PC1 values denote silent, late-replicating B compartments (Fig. 2b, c)[45]. Comparative PC1 analysis showed that although the majority of the bins in *Rif1*−/− shLacZ cells underwent minor changes in their PC1 values relative to shLacZ cells, a few bins did shift substantially, including PC1 sign flips in both directions, which gave a more dispersed appearance to the distribution relative to control cells (Fig. 2b, right). Visualization of PC1 profiles in various genomic regions confirmed that small changes in PC1 occurred in many locations in *Rif1*−/− shLacZ cells, but importantly, that these did not always correlate with the changes in RT (Fig. 2c). However, the magnitude of PC1 changes and the frequency of compartment switching (PC1 sign flips) is considerably milder than that reported in other cell types where loss of RIF1 led to substantial compartment switching[7,35]. In comparison, PC1 values in shMcm6 cells were comparable to those in shLacZ cells (Fig. 2b, left and Fig. 2c)[13].

We also generated PC1 versus RT density contour plots which allowed us to simultaneously compare the changes in compartmentalization and RT within the same set of genomic 20 kb bins. The density contour distribution profile in *Rif1*−/− shLacZ cells showed a marked shift of early RT bins towards delayed RT but without major changes in their PC1 values (Fig. 2d). This implies that the roles of RIF1 in RT and genome architecture are separable in B cells, as they are in other cells[7,35]. We grouped 20 kb genomic regions into 50 compartment categories based on their PC1 values and computed the average interaction frequency between all combinations of bins. The results are visualized as saddle plots (Fig. 2e). Normally, compartments interact preferentially with other compartments of the same type, as seen by the strong A-A or B-B interactions in shLacZ cells (Fig. 2f). *Rif1*−/− shLacZ cells showed a gain of A-B inter-compartmental interactions with a corresponding decrease in A-A and B-B contacts (Fig. 2f). These changes can be appreciated in the fold-change saddle plots showing an increase of contacts between regions of highest PC1 and lowest PC1 in *Rif1*−/− shLacZ cells (Fig. 2f). Thus, RIF1 plays a role in maintaining normal compartmentalization in B cells by preventing the mixing of A and B compartments. Importantly, this compartmentalization phenotype was observed in mESCs and hESCs where, in contrast to B cells, loss of RIF1 caused a severe deregulation of the RT program[7,35]. This suggests that the changes in compartmentalization in *Rif1*−/− cells are conserved between diverse cell lineages, but that these are unrelated to the changes in RT. Moreover, in shMcm6 cells, where RT is globally disrupted to a similar degree as in *Rif1*−/− mESCs and hESCs[7,13,35], we observed relatively minor changes in compartmentalization, as seen by the mild increase in A-A contacts and a similar decrease in B-B contacts (Fig. 2e, g).

In sum, the uncoupling of RT from genome organization in B cells and other cells[7,13,35] leads us to conclude that although RIF1 contributes to the normal spatial separation of A and B compartments in B cells, it's role in regulating early RT is unlikely to be directly linked to these structural changes.

## RIF1 is predominantly located in early-replicating transcribed chromatin in B cells

We next investigated the genomic occupancy of RIF1 in B cells. We performed chromatin immunoprecipitation (ChIP) from primary, mature splenic B cells derived from the *Rif1*FH/FH mouse line wherein RIF1 is endogenously tagged at its N terminus with the Flag and Hemagglutinin (HA2) epitopes[31,36]. RIF1 chromatin occupancy was determined via ChIP-sequencing (ChIP-seq) with an anti-HA antibody in *Rif1*FH/FH and *Rif1*+/+ cells (Supplementary Fig. 3a).

Visual analysis on the genome browser revealed RIF1 binding predominantly in early replicating domains in B cells (Fig. 3a). Comparison with ChIP-seq profiles from *Rif1*FH/FH mouse embryonic fibroblasts (MEFs)[36] showed that RIF1 was localized to broad domains in late-replicating regions in MEFs, whereas in B cells, enrichments were mostly in early-replicating domains (Fig. 3b). Moreover, RIF1 binding was observed across all autosomes with no evidence of selective association with specific chromosomes (Supplementary Fig. 3b, c).

We next identified peaks of RIF1 occupancy[46] using the MACS2 peak-calling software, which identifies peaks as regions of narrower and relatively strong enrichments over background[47]. However, RIF1 is also known to occur in RADs, which are broader domains of relatively lower signal enrichment. Hence, we used the EDD software which detects significant enrichments over broad genomic regions and has been used previously to identify RADs[36,47]. Only peaks that did not overlap with RADs were defined as peaks for further analyses. This yielded 16,043 and 862 RIF1 peaks in primary B cells and MEFs, respectively, and 289 and 332 RADs in primary B cells and MEFs, respectively. Thus, almost all (98.8%) RIF1-binding sites in B cells manifest as peaks of RIF1 enrichment.

Peaks in both cell types were enriched in early-replicating regions (Fig. 3c, d). Specifically, ~83% of peaks in MEFs and ~70% of peaks in primary B cells overlapped with transcription start sites (TSSs) or gene bodies (Supplementary Fig. 3d). RADs were mostly located in gene bodies or intergenic regions in both cell types (Supplementary Fig. 3d); however, RADs in MEFs were mostly late-replicating, as described[36], whereas in B cells, they were largely early-replicating (Supplementary Fig. 3c, d).

We next determined the overlap of RIF1 peaks in primary B cells with nascent transcription measured by precision run-on sequencing (PRO-seq)[48], chromatin accessibility measured by assay for transposase-accessible chromatin (ATAC-seq)[49], and ChIP-seq of histone H3 acetylated at lysine 27 (H3K27Ac), a mark of active TSSs and enhancers, and histone H3 trimethylated at lysine 9 (H3K9me3) a mark of silent heterochromatin. The data were used to create three sets of RIF1 peaks using k-means clustering, and subsequently, each cluster was ordered based on RIF1 density as shown in the heatmaps in Fig. 3e. Cumulative density histograms were also generated for each cluster, shown above the heatmaps (Fig. 3e). RIF1 intensity was similar in clusters 1 and 2 and slightly weaker in cluster 3. (Fig. 3e). Clusters 1 and 2 were marked by strong and comparable enrichments of nascent transcription signals and bimodal distributions of H3K27Ac around the peak center. Moreover, both clusters 1 and 2 showed strong ATAC-seq signal enrichments at the peak center, although cluster 2 was

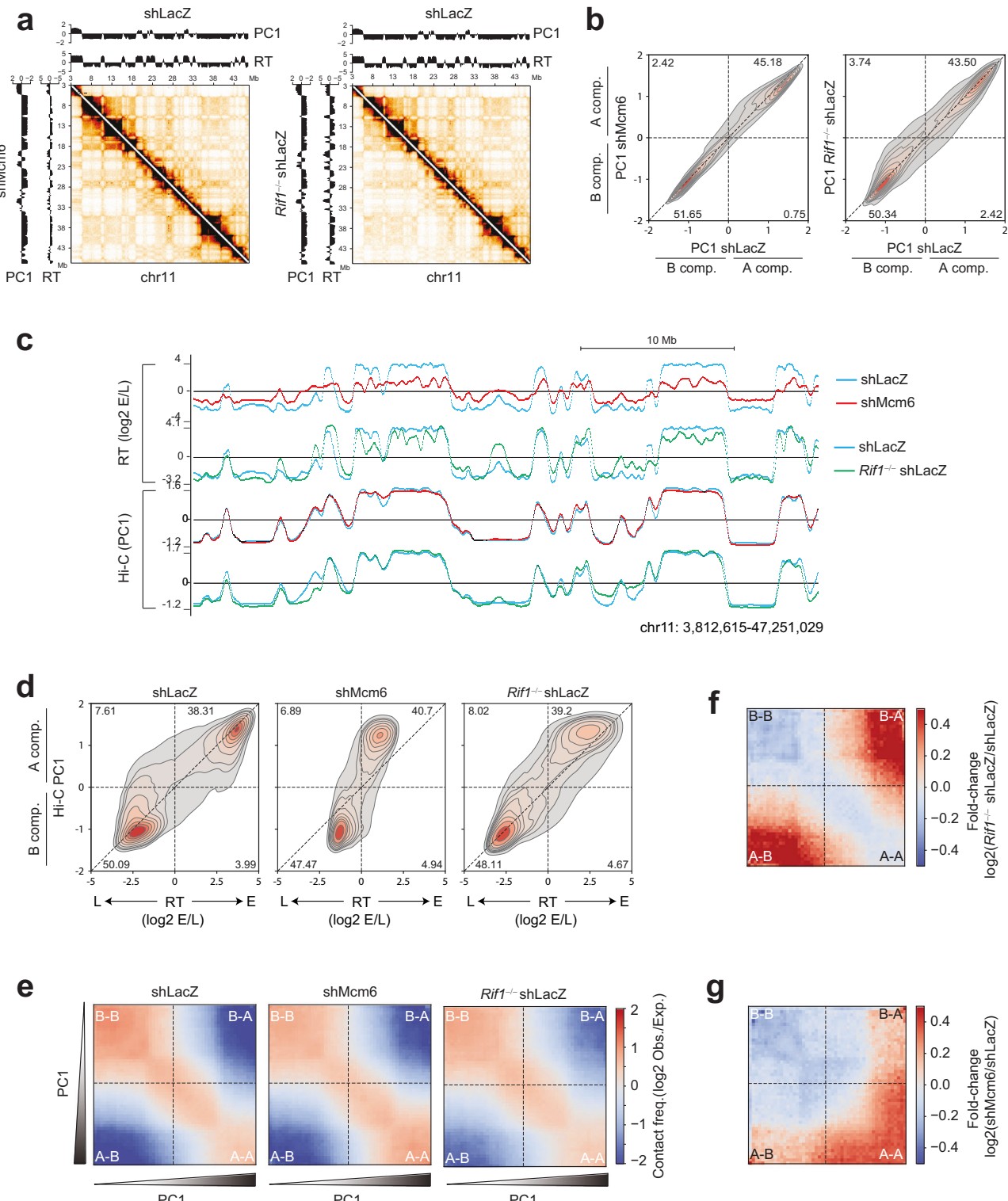

**Fig. 2 | RIF1 has a minor role in B cell genome architecture. a** Left: Hi-C contact matrix of chromosome 11 showing shLacZ contacts (above the diagonal) and shMcm6 contacts (below the diagonal) in CH12 cells. The PC1 compartment signals (Hi-C eigenvector eigenvalues) and RT (log2 E/L) tracks are shown above (for shLacZ) and on the left (for shMcm6) for each matrix. Right: Same as before but comparing shLacZ and *Rif1*[−/−] shLacZ cells. **b** Density-contour plot of PC1 compartment signals per 20 kb genomic bin. The left plot compares the PC1 signals in shLacZ and shMcm6 CH12 cells, and the right plot compares PC1 values between shLacZ and *Rif1*[−/−] shLacZ cells. The numbers within the plots are the percentage of bins in that quadrant. **c** A representative UCSC genome browser view comparing Hi-C PC1 profiles with RT (log2 E/L) profiles in shLacZ, shMcm6 and *Rif1*[−/−] shLacZ CH12 cells. **d** PC1 versus RT density-contour plots in 20 kb genomic bins to compare changes in compartmental identities (PC1) with RT in shLacZ, shMcm6 and *Rif1*[−/−] shLacZ CH12 cells. **e** Saddle plot from shLacZ, shMcm6 and *Rif1*[−/−] shLacZ CH12 cells showing long-range (>2 Mb) intra-chromosomal contact enrichments between bins of varying compartment signal strength (PC1). The values were computed from 20 kb KR-normalized contact matrices. **f, g** Fold-change saddle plots highlighting the changes in compartmentalization in *Rif1*[−/−] shLacZ cells (**f**) or shMcm6 cells (**g**) relative to shLacZ cells.

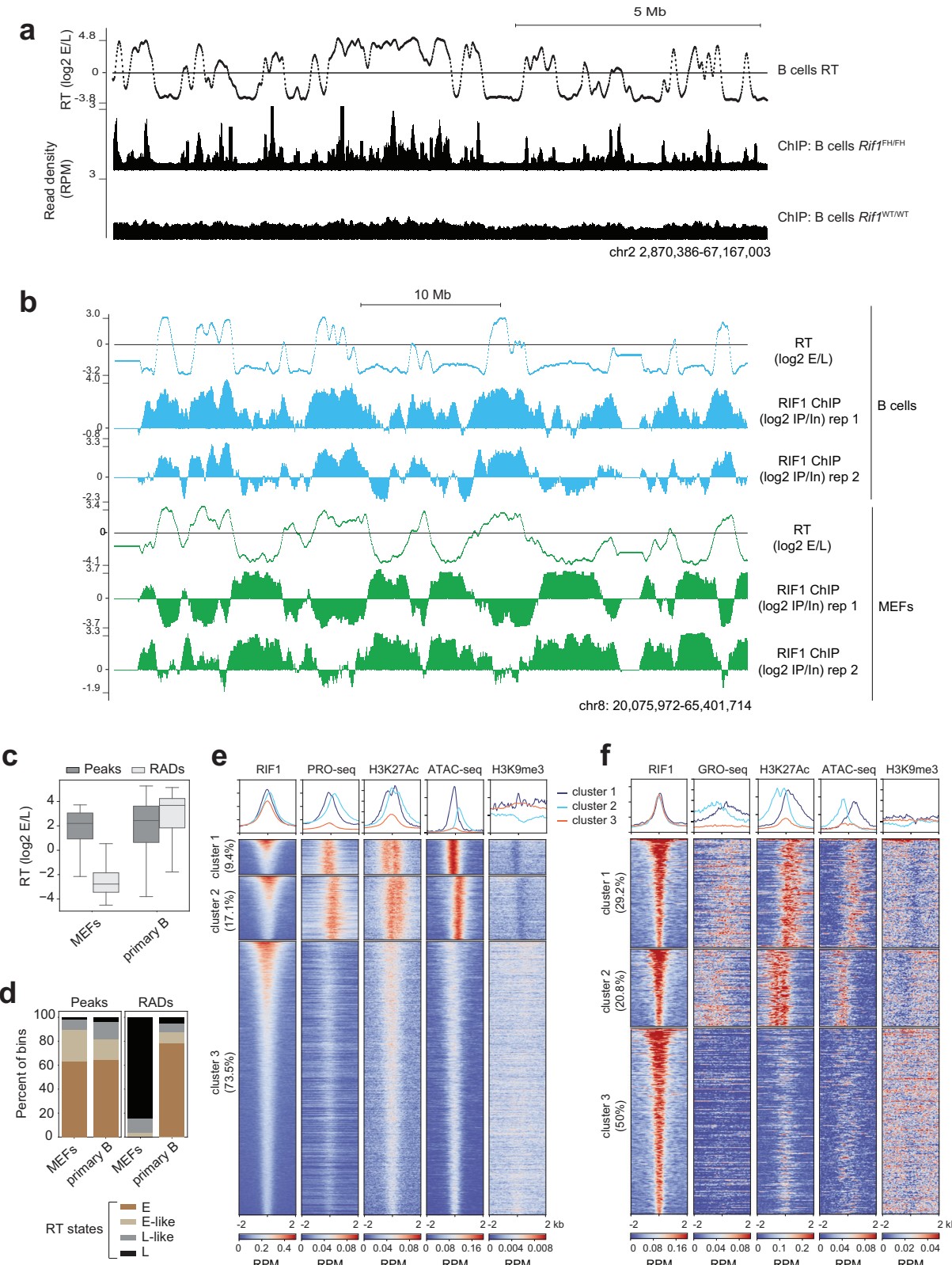

associated with lower ATAC-seq signal (Fig. 3e). These signatures are indicative of active TSSs, and accordingly, 63% of cluster 1 sites and 72% of cluster 2 sites were in TSSs or gene bodies (Supplementary Fig. 3e). Based on the location of the RIF1 peaks relative to H3K27Ac and ATAC-seq signals in clusters 1 and 2, which is best gauged from the cumulative histograms, we infer that RIF1 is bound upstream of the TSSs at these sites (Fig. 3e).

In contrast, cluster 3 in primary B cells, which contained the majority of RIF1 peaks (73.5%), were in regions with very low levels of transcription, H3K27Ac and ATAC-seq signal (Fig. 3e). Most of these sites were intergenic (33.7%) or in gene bodies (52.6%) of lowly transcribed or silent genes (Supplementary Fig. 3e). H3K9me3 signals were either absent (clusters 1 and 2) or very low (cluster 3) in the vicinity of RIF1 peaks (Fig. 3e), in line with RIF1 peaks being mostly

**Fig. 3 | RIF1 localizes predominantly to active chromatin in activated murine B cells. a** A representative UCSC genome browser view of RIF1 occupancy in murine *Rif1*^FH/FH primary, activated splenic B cells. ChIP-seq was performed with an anti-HA antibody and *Rif1*^WT/WT cells were used as a negative control. The RT track from WT, primary B cells provides a reference for early and late-replicating domains. **b** UCSC browser snapshot comparing ChIP enrichments of RIF1 in primary B and MEFs from *Rif1*^FH/FH mice. To allow direct comparison, ChIP signal was normalized to input signal and the ratio (ChIP/Input) tracks are shown. RT tracks are from WT cells. **c** Box plots comparing RT values of RIF1 peaks and RADs in MEFs and primary B cells. 16,043 peaks and 289 RADs were identified in primary B cells and 862 peaks

and 332 RADs were called in MEFs. The box bounds represent the 1st and 3rd quartile of the data distribution, the horizontal line represents the median and whiskers represent the 1st and 3rd quartile ± 1.5 times the interquartile distance. **d** Peaks and RADs were called as in C above and classified into the four RT states described in Fig. 1d. **e** Heatmap analysis of the chromatin locale in a 4 kb window surrounding RIF1 peaks called in *Rif1*^FH/FH cells and divided into three groups via k-means clustering, as described in the text. The heatmap is centered on the RIF1 peak summit and ordered by decreasing RIF1 read density (reads per million, RPM). The percentage of total peaks within each cluster is indicated. All datasets are from primary, activated wild-type B cells. **f** As in **e** above but for RIF1 peaks called in MEFs.

early-replicating (Fig. 3c, d). Importantly, within all clusters, RIF1 occupancy did not correlate with the levels of transcription, H3K27Ac or accessible chromatin.

We also performed the same analysis with k-means clustering of RIF1 peaks in MEFs. Here, all three clusters had similar RIF1 intensities (Fig. 3f). Clusters 1 and 2 were comparable with regards to transcription (measured here using global run-on sequencing, GRO-seq[50]) and high enrichments of H3K27Ac and ATAC-seq signals, suggesting that these peaks are located in active chromatin regions (Fig. 3f). Indeed, ~92% of cluster 1 and ~75% of cluster 2 sites overlapped TSSs or gene bodies (Supplementary Fig. 3e). In comparison, cluster 3 peaks were associated with very weak or no transcription or ATAC-seq signals (Fig. 3f) with ~80% of these associated with TSSs or gene bodies of weakly expressed or silent genes (Supplementary Fig. 3e). In agreement with the fact that most RIF1 peaks in MEFs are in early-replicating regions (Fig. 3c, d), H3K9me3 was poorly enriched around RIF1 peaks (Fig. 3f).

In sum, these analyses reveal that RIF1 predominantly occupies early-replicating domains in B cells suggesting that RIF1 promotes their early replication via direct association. However, within these domains, RIF1 binds to both transcribed and non-transcribed regions, implying that the genomic occupancy of RIF1 is not solely determined by transcriptional strength or the degree of accessible chromatin. We note that the lack of RIF1 in late RT domains in B cells provides a plausible explanation for the relatively weak effects on late RT in *Rif1*^−/− B cells as compared to other *Rif1*^−/− cell lines, which harbor RADs in late RT domains.

## RIF1 is not a major regulator of origin activity in B cells

The localization of RIF1 in active chromatin coupled with the delayed replication of these domains in *Rif1*^−/− cells led us to investigate whether RIF1 played a role in promoting origin firing in activated B cells. To address this, we measured origin activity with short nascent strand sequencing (SNS-seq) in WT and *Rif1*^−/− CH12 cells (Fig. 4a and Supplementary Fig. 4). SNS-seq maps the location and relative usage of origins (termed replication initiation sites; ISs) in a population of cells by quantifying the levels of nascent leading strands[51,52]. As described in our previous work[13], ISs and initiation zones were defined based on peak-calling and peak-clustering, respectively, and only ISs within initiation zones were used for analysis. SNS-seq identified 52,193 ISs in WT and *Rif1*^−/− cells (Fig. 4a).

To quantify the changes in origin efficiency, we calculated fold-changes of IS read densities (log2 WT/*Rif1*^−/−). Only ~5% of ISs showed >1.5-fold change in read density upon loss of RIF1 compared to ~40% in shMcm6 cells (Fig. 4b), indicating that RIF1 is not a major regulator of replication origin activation in B cells. However, because RIF1 has been previously implicated in origin licensing in human 293 cells[43], we asked whether decreased licensing could explain why a minor subset of ISs were downregulated in *Rif1*^−/− cells. We performed MCM5 ChIP-qPCR at five of the most downregulated ISs from SNS-seq analyses and compared the results with MCM5 occupancy at four ISs that were unchanged in *Rif1*^−/− cells (Supplementary Fig. 4b). We found a significant decrease in MCM5 enrichment at all downregulated ISs, but not at the unchanged ISs (Supplementary Fig. 4c). Therefore, although

RIF1 does not regulate replication origin activity at the vast majority of origins in B cells, a small subset of origins requires RIF1 for optimal firing efficiency.

To gain further insight into the locations of the deregulated origins, we generated heatmaps of IS read densities, which provide a qualitative assessment of IS location relative to chosen genomic features. The heatmaps span a 4 kb window centered at the IS peak summit and ordered by the IS density in WT cells (Fig. 4c). To visualize how IS density correlated with various chromatin features in the genomic neighborhood, we also generated heatmaps showing the densities of DNase hypersensitive sites (DHSs) which mark accessible chromatin at active promoters and enhancers, H3K36me3, which is enriched in the bodies of transcriptionally active genes, and the repressive heterochromatin mark, H3K9me3 (Fig. 4c). In shLacZ cells, the most active ISs were embedded in active chromatin and gene bodies whereas the least active ISs were associated with H3K9me3 (Fig. 4c). We observed a trend wherein the most downregulated ISs (top of the heatmap) were located in active chromatin and the most upregulated origins (bottom of the heatmap) were embedded in silent chromatin (Fig. 4c). These trends were reminiscent of what we previously reported in shMcm6 B cells[13] suggesting that, in B cells, RIF1 may modulate the activity of a small subset of origins by regulating the recruitment of MCM proteins or the assembly of the pre-RC.

To systematically quantify the changes in origin efficiency, we ranked ISs based on their fold-changes (log2 WT/*Rif1*^−/−) from highest to lowest and created five equal classes (quintiles) such that class 1 contained the most downregulated ISs in *Rif1*^−/− cells and class 5 contained the most upregulated ISs in *Rif1*^−/− cells. We next generated violin plots displaying the read density of ISs (Fig. 4d) or RT values (Fig. 4e) in WT and *Rif1*^−/− cells within each IS class. The results showed that in WT cells, class 1 ISs were the most active (Fig. 4d) and were mostly early-replicating (Fig. 4e) whereas class 5 ISs were the least active with the majority being late-replicating (Fig. 4d, e). Importantly, class 5 IS densities in *Rif1*^−/− cells were comparable to class 1 IS densities in WT cells, suggesting that the upregulated origins in *Rif1*^−/− cells (class 5), which are normally the weakest, fire at similar efficiencies as the most active origins in WT cells (class 1) (Fig. 4d).

We conclude that although RIF1 is not a major regulator of origin activity in B cells, it is required for the optimal activity of a few early replicating origins.

## RIF1 and MCM complexes act in a complementary manner to regulate the B cell RT program

The differing RT phenotypes in *Rif1*^−/− and shMcm6 cells led us to hypothesize that they may function in a non-epistatic manner to drive early replication in B cells. To address this, we infected *Rif1*^−/− CH12 cells with lentiviruses expressing shMcm6 (*Rif1*^−/− shMcm6) or shLacZ (*Rif1*^−/− shLacZ) (Supplementary Fig. 5a) and performed Repli-seq (Supplementary Fig. 1a). Of note, the converse experiment, that is, depletion of RIF1 in shMcm6 cells, was precluded by the fact that the viability of shMcm6 cells was severely compromised upon viral infection with *Rif1*-specific sgRNAs or shRNAs, in line with the observation that cells with limiting MCM proteins are sensitive to stress[53–55].

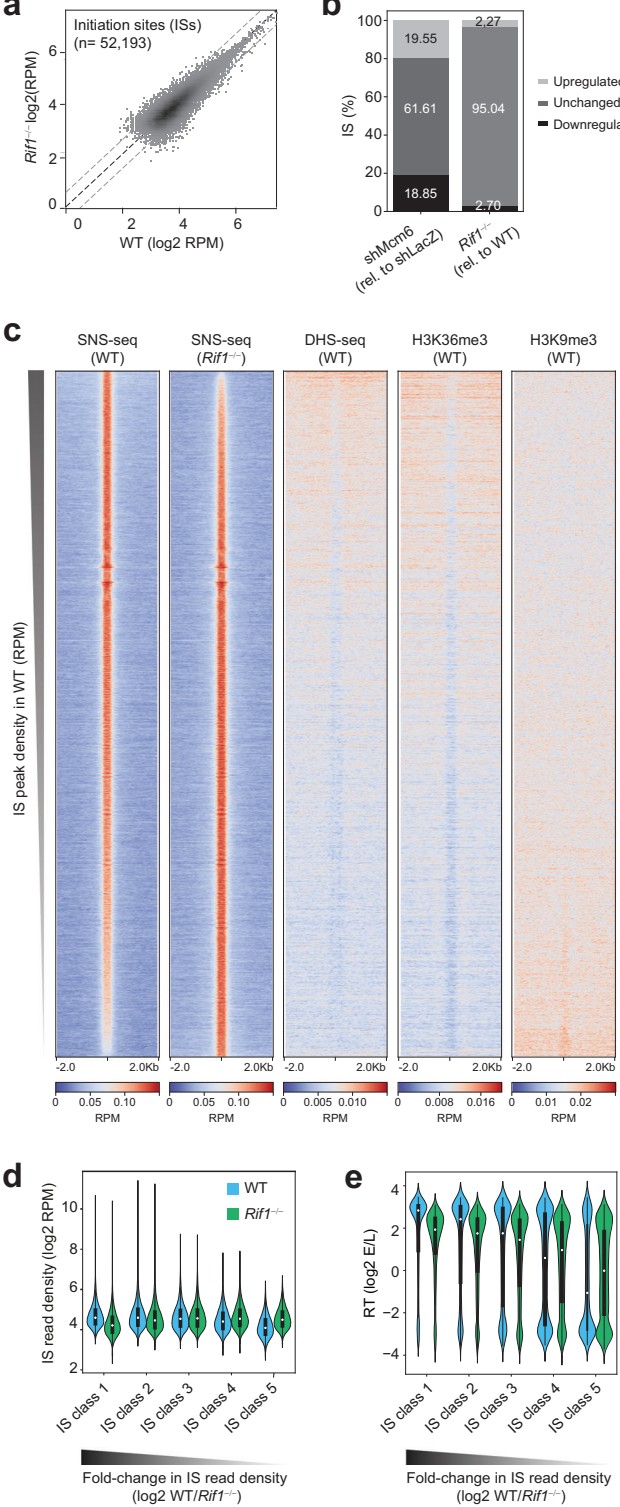

**a** Scatter plot of IS density in log2 RPM (reads per million) in WT and *Rif1⁻/⁻* CH12 cells identified from SNS-seq. Data were obtained from two replicates (*n* = 2), as shown in Supplementary Fig. 4a. **b** Bar plots showing the percentage of upregulated (1.5-fold), unchanged or downregulated (1.5-fold) ISs in shMcm6 cells relative to shLacZ cells and in *Rif1⁻/⁻* cell relative to WT cells. The shMcm6 and shLacZ data are from our previous study (ref. 13). **c** Heatmaps spanning a 4 kb region centered on the IS peak summit providing an overview of the location of ISs relative to regions of DNase hypersensitivity (DHS)-seq and histone modifications (H3K36me3 and H3K9me3). All heatmaps display read densities as RPM. Within each heatmap, the ordering is based on decreasing SNS-seq fold-change (*Rif1⁻/⁻*/WT). **d** ISs in CH12 cells were split into five equal classes (quintiles) based on their fold-change of read densities (log2 *Rif1⁻/⁻*/WT) such that class 1 contained the most downregulated ISs and class 5 contained the most upregulated ISs, respectively, in *Rif1⁻/⁻* cells. The violin plots show the IS read density within each class in WT and *Rif1⁻/⁻* cells. Number of IS per class: class 1 = 7827, class 2 = 7819, class 3 = 7824, class 4 = 7828, class 5 = 7817. The violins represent kernel density estimates of the value distribution. The inner boxes and whiskers represent the same data as a boxplot where the dot represents the median, the minimum and maximum of the box represent the 1ˢᵗ and the 3ʳᵈ quartile, and the minimum and maximum of the whiskers are the 1ˢᵗ and 3ʳᵈ quartile ± 1.5 times the interquartile distance. **e** ISs were classified into five classes as in **d**. The violin plots show the RT values within each IS class.

shMcm6 cells, many large early-replicating domains showed considerable fluctuation in the RT values resulting in a highly fragmented RT profile (Fig. 5d). However, this fragmentation was considerably lower in large late-replicating domains (Fig. 5d). The RT profiles in *Rif1⁻/⁻* shMcm6 cells were also marked by switching of RT signatures (both E to L and L to E) and loss of clear domain boundaries (Fig. 5d). These changes, and especially the global fluctuations of early RT values, result in *Rif1⁻/⁻* shMcm6 cells acquiring an RT signature distinct from shMcm6 cells (Fig. 5a, c). In sum, these findings reveal an additive effect of MCM depletion and loss of RIF1 on RT in B cells.

Despite the changes in RT, gene expression (Supplementary Fig. 5b, c and Supplementary Data 2), compartment identity (Fig. 5d and Supplementary Fig. 5d) and compartment strength (Fig. Supplementary 5e, f) were not majorly altered in *Rif1⁻/⁻* shMcm6 cells relative to *Rif1⁻/⁻* shLacZ cells, although slight gains in A-B, B-A and A-A interactions were observed, akin to the changes seen in shMcm6 cells (Supplementary Fig. 5f, compare with Fig. 2g). However, the major changes in RT led to a distinct pattern of PC1 versus RT profiles in in *Rif1⁻/⁻* shMcm6 cells relative to in *Rif1⁻/⁻* shLacZ cells (Supplementary Fig. 5e). Thus, the deregulation of the RT program by MCM depletion does not majorly impact genome architecture in normal (Fig. 2) or in *Rif1⁻/⁻* cells.

To determine whether the increased deregulation of RT in *Rif1⁻/⁻* shMcm6 cells was associated with changes in underlying replication origin activity, we performed SNS-seq in *Rif1⁻/⁻* shLacZ and *Rif1⁻/⁻* shMcm6 CH12 cells (Fig. 6a and Supplementary Fig. 4). SNS-seq identified 129,937 ISs in *Rif1⁻/⁻* shLacZ and *Rif1⁻/⁻* shMcm6 (Fig. 6a). ISs in active chromatin were downregulated in *Rif1⁻/⁻* shMcm6 cells and this was accompanied by an upregulation of IS activity in heterochromatin (Fig. 6b), consistent with the phenotype we reported previously in shMcm6 cells[13]. To quantitatively assess changes in IS densities, we classified ISs into quintiles based on fold-changes in read densities (*Rif1⁻/⁻* shLacZ/*Rif1⁻/⁻* shMcm6) and analyzed the distribution of IS read densities (Fig. 6c) and RT values (Fig. 6d) within the IS classes. The most downregulated ISs (class 1) were the most active in *Rif1⁻/⁻* shLacZ and were predominantly early replicating, whereas the most upregulated ISs (class 5) were the least active in *Rif1⁻/⁻* shMcm6 cells and were mostly late-replicating (Fig. 6c, d). SNS-seq genomic profiles showed a characteristic downregulation of origin activity in A compartments accompanied by upregulation in B compartments in both shMcm6 and *Rif1⁻/⁻* shMcm6 cells (Fig. 6e).

Repli-seq revealed that there was a further loss of early and late RT domain distinction in *Rif1⁻/⁻* shMcm6 cells compared to *Rif1⁻/⁻* shLacZ cells or shMcm6 cells, with a shift of both E and L RT values towards zero (Fig. 5a). An HMM-based classification of RT values, as in Fig. 1d, showed that E bins replicated later, and L bins replicated earlier in *Rif1⁻/⁻* shMcm6 compared to shMcm6 cells (Fig. 5b). Direct comparison of RT values in 20 kb genomic bins showed that this exacerbation of the RT phenotype in *Rif1⁻/⁻* shMcm6 relative to *Rif1⁻/⁻* shLacZ and shMcm6 cells was observed in most of the RT bins (Fig. 5c). Additionally, in *Rif1⁻/⁻*

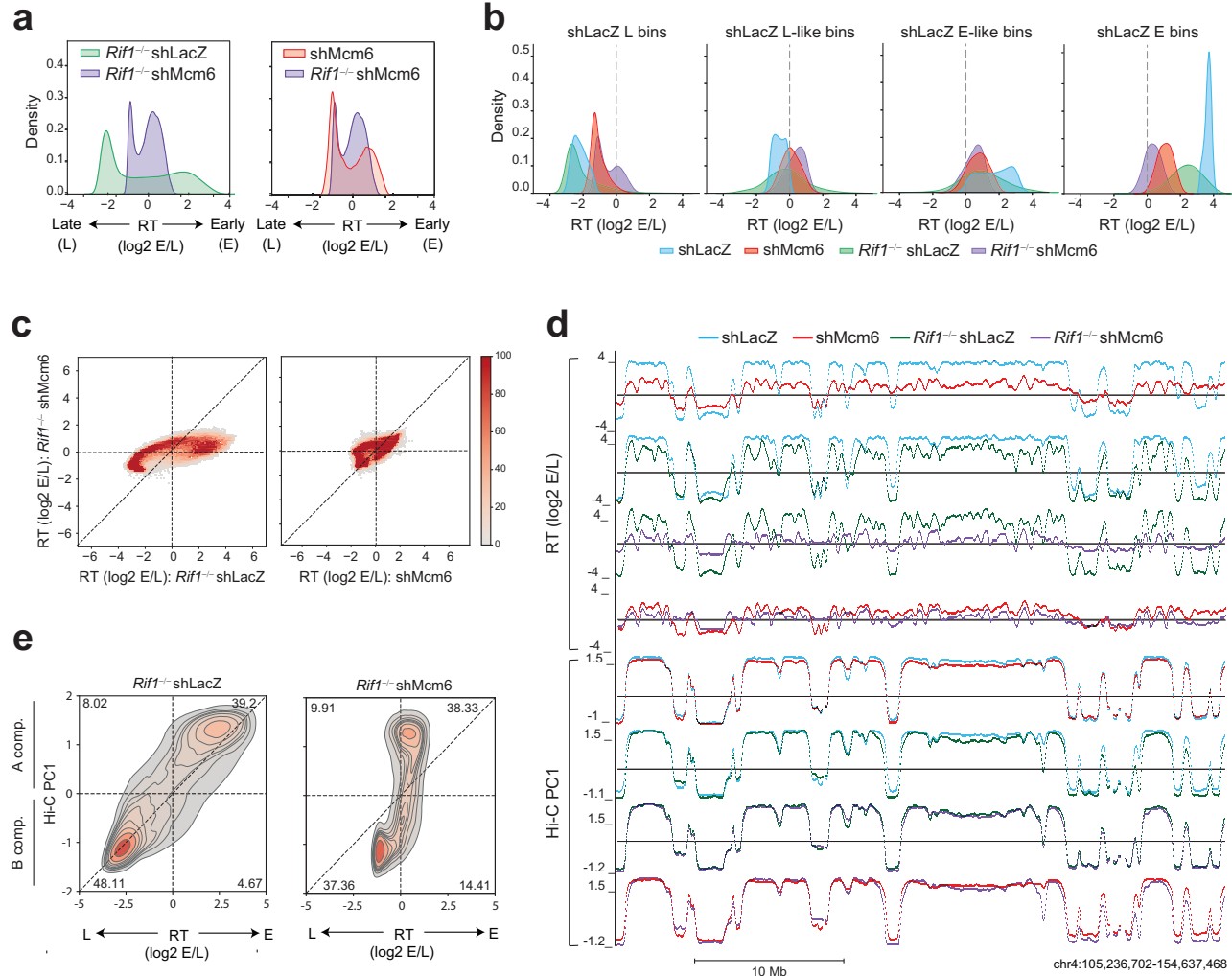

**Fig. 5 | RIF1 and MCM complexes act in an additive manner to regulate early RT in B cells. a** RT histogram comparing *Rif1⁻/⁻* shLacZ and *Rif1⁻/⁻* shMcm6 CH12 cells. **b** Identical to Fig. 1e showing how HMM-based RT states called in shLacZ CH12 cells change in the other three conditions. **c** Comparison of RT values in 20 kb genomic bins between *Rif1⁻/⁻* shMcm6 and *Rif1⁻/⁻* shLacZ (left) and between *Rif1⁻/⁻* shMcm6 and shMcm6 CH12 cells (right). **d** A representative UCSC genome browser view comparing Hi-C PC1 profiles with RT (log2 E/L) profiles in shLacZ, shMcm6, *Rif1⁻/⁻* shLacZ and *Rif1⁻/⁻* shMcm6 CH12 cells. **e** PC1 versus RT density-contour plots in 20 kb genomic bins comparing the changes in either feature in *Rif1⁻/⁻* shLacZ and *Rif1⁻/⁻* shMcm6 CH12 cells.

Collectively, we infer that the additional delay in early replication in *Rif1⁻/⁻* shMcm6 cells relative to *Rif1⁻/⁻* shLacZ cells is likely due to the further decrease in the efficiency of early-firing origins.

We conclude that MCM proteins and RIF1 regulate RT in a complementary and non-epistatic manner in B cells.

### Within early RT domains, the RT of highly transcribed regions is most sensitive to the depletion of MCM complexes and, to a weaker extent, the loss of RIF1

A closer examination of the RT profiles revealed that the extensive fragmentation seen in *Rif1⁻/⁻* shMcm6 cells was also visible to a lesser extent in shMcm6 and to the weakest extent in *Rif1⁻/⁻* shLacZ cells (Fig. 5c, d). Furthermore, we observed a high degree of similarity in the fragmentation patterns between the different conditions. This suggested that these were not random fluctuations of the RT signals but reflected an underlying mechanism supporting early replication that is reliant on MCM proteins and, to a weaker extent, RIF1. Therefore, we investigated whether the levels of nascent transcription within genomic RT bins could explain these altered RT profiles. This reasoning was based on the observations that fragmentation was typically seen in early RT domains where most transcribed genes are located, and secondly, that in *Rif1⁻/⁻*, shMcm6 and *Rif1⁻/⁻* shMcm6 cells, early origins

in active chromatin were downregulated despite there being no major changes in transcription.

To address this, we divided the genome into 20 kb bins and extracted all early-replicating E bins in shLacZ cells using the HMM-based approach described in Fig. 1c. Within this group, we defined transcribed bins as those having a PRO-seq density (RPM) ≥ 10 and these were divided into three equal sub-groups (tertiles) based on their RPM values, termed High (199–3358 RPM), Medium (78–198.9 RPM) and Low (10–77.9 RPM) (Supplementary Fig. 5g). The remaining early-replicating bins were termed Untranscribed (0–9.9 RPM) (Supplementary Fig. 5g). To compare changes in RT values, we generated RT heatmaps for the four transcription-based groups and ranked each of them by decreasing WT PRO-seq read density such that, effectively, the entire set of bins were ranked from highest to lowest PRO-seq density in shLacZ cells (Fig. 7a). In addition, we generated density plots to visualize the differences in the distribution of RT values between the four groups in all experimental conditions (Fig. 7b).

In shLacZ cells, all four transcription-based groups showed nearly identical distributions of RT values, indicating that, normally, transcribed and non-transcribed regions have similar probabilities of origin activation and early replication (Fig. 7a, b). The High and Medium

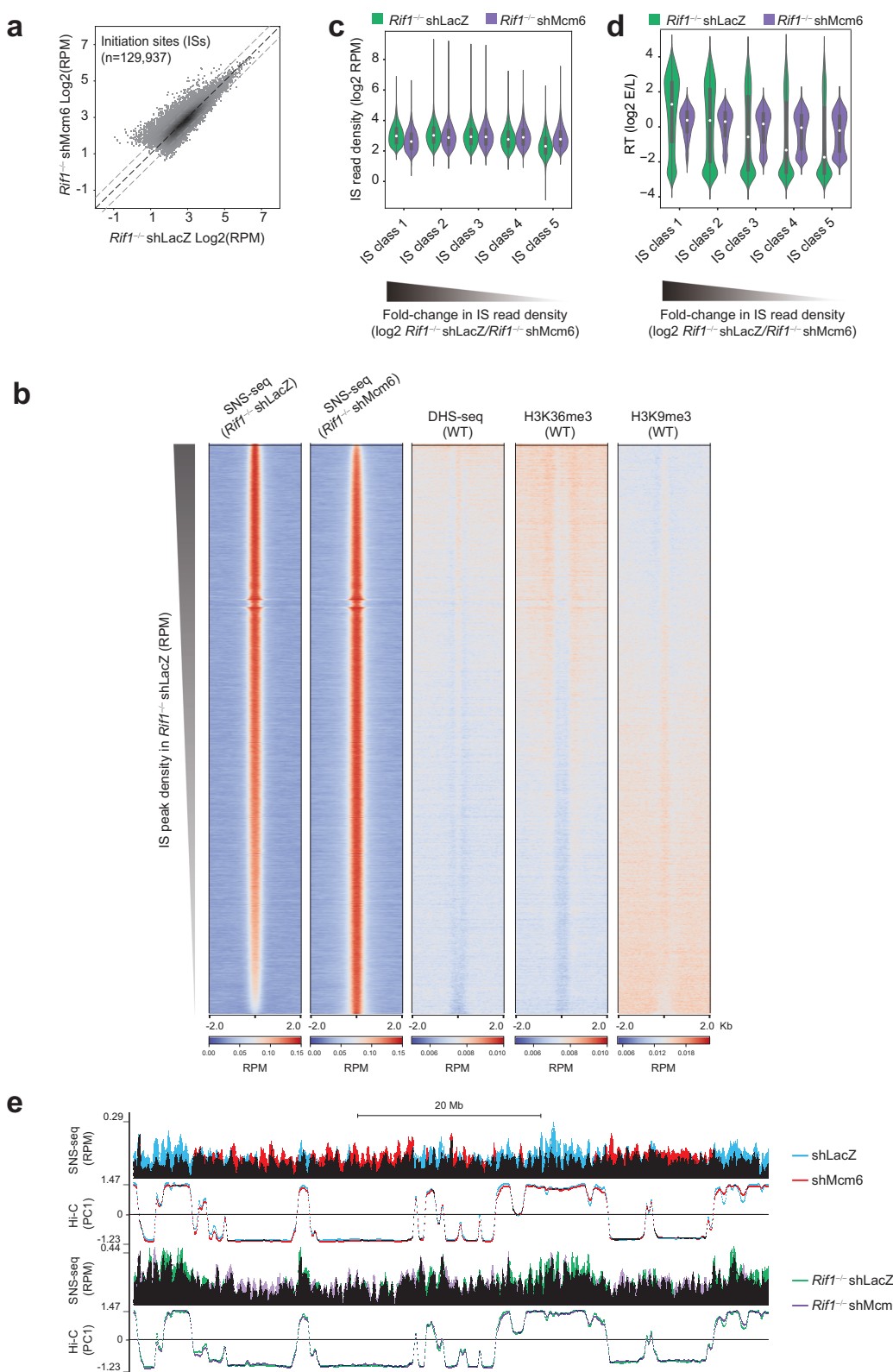

groups appeared very similar in terms of the RT values between bins (Fig. 7a) and the overall distribution profiles (Fig. 7b). However, in shMcm6 and $Rif1^{-/-}$ shMcm6 cells, the High and Medium transcribed bins showed the strongest delays in RT whereas the Untranscribed group showed the least delays (Fig. 7a, b). These differences were observed across all bins (Fig. 7a). The shift in RT between the High and Untranscribed groups was ~2-fold (log2 RT ~1), as measured by the

distance between the modes of the distributions (marked by dashed lines in Fig. 7b). The lowly transcribed regions had an intermediate distribution of RT values between the High/Medium and Untranscribed groups in both shMcm6 and $Rif1^{-/-}$ shMcm6 cells (Fig. 7a, b). Thus, the strongest delays in RT are associated with regions harboring higher levels of transcription. In $Rif1^{-/-}$ shLacZ cells, all groups had a broad spread of RT values with similar distribution profiles (Fig. 7a, b).

**Fig. 6 | MCM depletion in _Rif1_<sup>−/−</sup> cells leads to a further deregulation of early origin activity. a** Scatter plot of IS density in log2 RPM (reads per million) in _Rif1_<sup>−/−</sup> shLacZ and _Rif1_<sup>−/−</sup> shMcm6 CH12 cells identified from SNS-seq. Data were obtained from two replicates (_n_ = 2), as shown in Supplementary Fig. 4a. **b** Heatmaps, generated as described in Fig. 4b, providing a qualitative assessment of the distribution of IS peaks relative to their local chromatin environment. **c** ISs in CH12 cells were split into five equal classes (quintiles) based on their fold-change of read densities (log2 _Rif1_<sup>−/−</sup> shMcm6/_Rif1_<sup>−/−</sup> shLacZ) such that class 1 contained the most downregulated ISs and class 5 contained the most upregulated ISs, respectively, in _Rif1_<sup>−/−</sup> shMcm6 cells. The violin plots show the IS read density within each class in _Rif1_<sup>−/−</sup> shLacZ and _Rif1_<sup>−/−</sup> shMcm6 cells. Number of IS per class: class 1 = 25,987, class

2 = 25,987, class 3 = 25,987, class 4 = 25,993, class 5 = 25,982. The violins represent the kernel density estimates of the value distribution. The inner boxes and whiskers represent the same data as a boxplot where the dot represents the median, the minimum and maximum of the box represent the 1<sup>st</sup> and the 3<sup>rd</sup> quartile, and the minimum and maximum of the whiskers are the 1<sup>st</sup> and 3<sup>rd</sup> quartile ± 1.5 times the interquartile distance. **d** ISs were classified into five classes as in C above. The violin plots show the RT values within each IS class. **e** Representative genomic snapshot of SNS-seq and HiC PC1 profiles. Data from shLacZ (blue) and shMcm6 (red) CH12 cells are overlaid with black being the overlap between them. _Rif1_<sup>−/−</sup> shLacZ and _Rif1_<sup>−/−</sup> shMcm6 CH12 tracks are in green and purple, respectively, with black indicating the overlap between them.

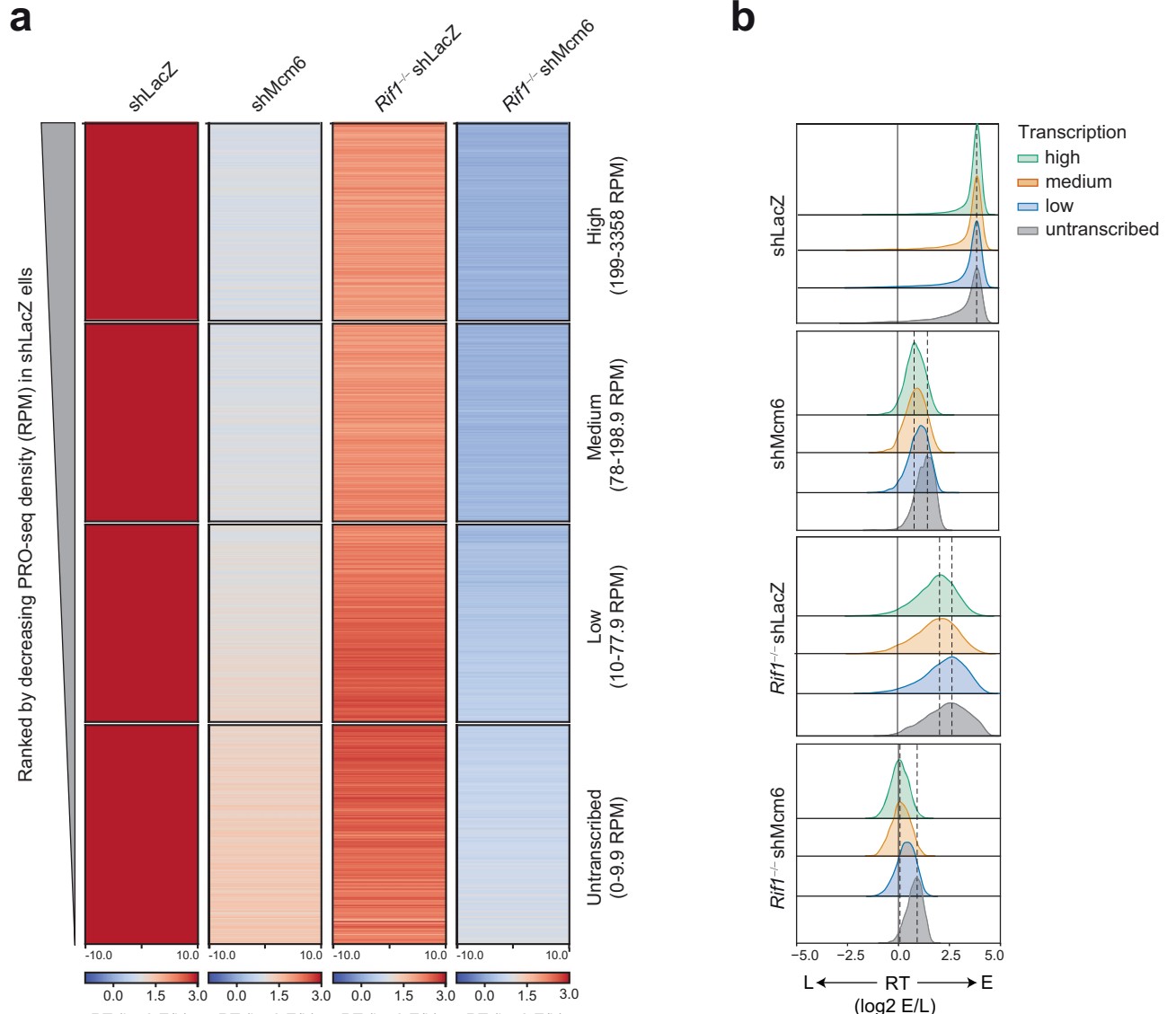

**Fig. 7 | Within early-replicating domains, the RT of highly transcribed regions is more sensitive to reduction in MCM proteins and loss of RIF1 than poorly transcribed regions. a** Heatmap of RT values in 20 kb E bins from shLacZ CH12 cells that were split into four groups based on PRO-seq RPM. The range of PRO-seq RPMs in each group is indicated on the left. All heatmaps are ranked from highest to lowest PRO-seq density such that the entire set of heatmaps is effectively ranked from highest to lowest PRO-seq density for all shLacZ E bins. **b** Density distribution of the RT (top) values in the four transcription-based groups from A above. The dashed black lines indicate the modes of the distributions in the High and Untranscribed classes.

However, a closer inspection of the heatmap and the modes of the distributions revealed a slightly greater delay in RT for the high and medium transcribed regions compared to the lowly transcribed and non-transcribed bins (Fig. 7b). Thus, the differential regulation of RT between transcribed and nontranscribed regions is mostly due to MCM complexes rather than RIF1.

Taken together, these results suggest that the RT of transcribed regions, and hence the underlying activity of early origins, is considerably more sensitive to the reduction of MCM proteins than to the loss of RIF1, which correlates with the magnitude of the fragmentation observed in the early RT domains (Fig. 5c, d). We conclude that the fragmented appearance of early RT profiles is due to underlying

transcription, with stronger delays correlating with higher levels of transcription. These results suggest that transcriptional strength negatively regulates the probability of early origin firing at active genes, and that this is normally overcome by a mechanism driven by the presence of a normal complement of MCM proteins with RIF1 making a minor contribution.

## Discussion

Our study reveals two layers of regulation within the RT program. First, a mechanism that specifically promotes early RT via the functional repurposing of RIF1. Second, the importance of normal levels of MCM proteins, with a minor role for RIF1, in ensuring early RT of transcribed regions.

Although the mechanism by which RIF1 regulates early RT in B cells remains to be deciphered, our data suggest that the RT phenotype derives primarily from RIF1-dependent modulation of the timing of origin activation with a minor contribution from RIF1-mediated origin licensing. The fact that RIF is a positive regulator of origin firing in B cells suggests that its early RT function is unlikely to be mediated via a mechanism involving the canonical PP1-mediated dephosphorylation of MCM4, which is a pathway that suppresses origin activity, although we do detect hyperphosphorylated MCM4 in *Rif1⁻/⁻* B cells. Instead, we propose that RIF1 could function via a PP1-based mechanism involving the dephosphorylation of other replication proteins. A precedent for this is the recent finding that PP2A- and PP4-mediated dephosphorylation of key initiation factors is essential for replication[56]. Accordingly, RIF1 may regulate the recruitment of replication initiation factors to the pre-RC at early origins, perhaps involving PP1 dephosphorylation, thereby influencing their firing time. A previous study in fission yeast found that loss of RIF1 affected the loading of Cdc45 to the pre-RC, a critical early step in origin activation[33]. It was proposed that loss of RIF1 may lead to increased Cdc45 loading at some late origins which would limit the number of Cdc45 molecules for activation of early origins resulting in delayed firing of the latter origins[33]. Thus, it is plausible that sequestration of rate-limiting replication initiation factors like Cdc45 by some late origins may contribute to the early replication phenotype of *Rif1⁻/⁻* B cells.

It has been reported that RIF1 can promote origin licensing by protecting ORC1 from phosphorylation-mediated degradation[43]. Also, RIF1 has been shown to interact with MCM complexes in activated murine B cells[38]. In agreement, we find a role for RIF1 in origin licensing, as judged by decreased origin activity and MCM5 chromatin occupancy. However, this appears to be a minor function of RIF1 in B cells since only 5% of ISs showed changes (1.5-fold) in activity in the absence of RIF1. We note, however, that SNS-seq may not be efficient at detecting weaker or less frequently used origins. Hence, we cannot exclude the possibility of RIF1 having a more prominent role in origin licensing than that revealed by SNS-seq. If so, it is plausible that even if the effects of RIF1 on licensing individual early origins is small, the cumulative effects across many such origins could explain, in part, the observed delays in early origin firing in *Rif1⁻/⁻* B cells.

Another insight from our study is the negative impact of transcriptional strength on origin firing time. But how might transcription negatively impact the efficiency of replication origins? There is evidence from in vitro studies that RNA polymerase (RNAP) complexes can push MCM complexes along DNA[57,58]. Indeed, MCM and ORC proteins were found to be relatively depleted in gene bodies[59] and Okazaki fragment sequencing studies have reported that replication initiation zones occur largely in the intergenic regions flanking transcribed genes[60]. A recent single-molecule imaging study, where replication-transcription encounters were reconstituted using purified proteins, may be instructive in this regard[58]. This study found that T7 RNAP could efficiently push DNA-bound ORC, OCCM (an intermediate of the pre-RC consisting of ORC, Cdc6, Cdt1 and a single MCM hexamer) and MCM double hexamers (the

configuration within the fully assembled pre-RC). However, whereas OCCM and double hexamers were rarely ejected by RNAP, ORC alone was frequently evicted by RNAP. Moreover, most of the ORC molecules repositioned by RNAP were unstable[58]. Since ORC loading is the first step in origin licensing[25], the inference is that the labile binding of ORC makes licensing in transcribed regions inherently less efficient than in non-transcribed regions, but that this is overcome by the association of MCM complexes, which minimizes the loss of single ORC complexes by RNAP.

Importantly, our data shows that, normally, the distribution of early RT values is similar between transcribed and non-transcribed regions. Hence, we propose that, in WT cells, the large pool of MCM complexes ensures that the loading of ORC is rapidly followed by the assembly of the OCCM and pre-RC such that the loss of single ORC complexes by RNAP is minimized. This ensures that both transcribed and non-transcribed regions have a similar probability of early origin activation and early replication in WT cells. When MCM complexes are limiting, licensing in both transcribed and non-transcribed regions is reduced. However, given the inhibitory effect of transcription on ORC stability, the formation of OCCM and pre-RC formation in highly transcribed regions will be inefficient, allowing for more eviction of ORC by RNAP. This will reduce early origin licensing efficiencies in highly transcribed chromatin to a greater degree than in non-transcribed chromatin, leading to our observation of a higher probability of early replication in nontranscribed regions relative to transcribed regions.

Given that RIF1 has not been previously implicated in regulating early replication, our study raises the question of why antigen-activated B cells have functionally repurposed RIF1 in this manner. Antigen-activated B cells are amongst the fastest proliferating cells in the body with a cell cycle duration of 6–8 h in vivo within germinal centers and 8–12 h in culture[61–63]. A unique feature of these cells is that they undergo extremely high rates of somatic hypermutation at the highly transcribed immunoglobulin genes, a necessary event in antibody maturation upon infection or vaccination. Genome instability is further elevated by the fact that somatic hypermutation also occurs at many other transcribed loci, including proto-oncogenes like *BCL6* and *MYC*[64–67], which can result in oncogenic translocations typical of most mature B cell cancers[68–70]. Moreover, the DNA repair pathways activated by somatic hypermutation result in single- and double-strand breaks as well as singe-strand patches, all of which are impediments for replication[71,72]. Therefore, these B cells proliferate in a highly genotoxic environment where replication stress is elevated. Under such conditions, it is plausible that B cells have functionally repurposed RIF1 as an additional mechanism to safeguard its genome by enforcing the early replication of active genes, which would help to decrease mutations and common fragile sites associated with late replication or delayed replication of long genes[11,14–20]. It is also possible that this function of RIF1 is masked in other cell types where the dominant mode of RT regulation is that of RIF1-mediated suppression of origins in late-replicating chromatin[7,34,35].

## Methods

### Cell culture

CH12 cells (clone F3)[73] were maintained in RPMI medium with 10% fetal bovine serum, glutamine, sodium pyruvate, Hepes and antibiotic/antimycotic mix. LentiX packaging cells were maintained in DMEM medium with 10% FBS and antibiotics. Where indicated, tautomycin (Santa Cruz Biotechnology, sc-200587) was added at a final concentration of 1 μM for 2 h.

### Mice

*Rif1*^FH/FH and *Rif1*^+/+ mice were described previously[31] and maintained on a C57BL/6 background. Mice were kept in a specific pathogen-free (SPF) barrier facility at a temperature of 20 ± 2 °C, humidity at 55% ±

15% and a 12 h:12 h light:dark cycle. Animals were maintained in small groups (4–5) or as breeding pairs in individually ventilated cages and had uninterrupted access to food and water. Male and female mice were used indiscriminately for experiments. All experiments were carried out in accordance with the European Union (EU) directive 2010/63/EU, and in compliance with Landesamt für Gesundheit und Soziales directives (LAGeSo, Berlin, Germany).

## Transfections and infections

LentiX packaging cells were transfected with shLacZ or shMcm6 expressing pLKO lentiviral vectors along with the packaging plasmid (delta 8.9) and the envelope plasmid (VSV-G)[13,39] using FuGene transfection reagent (Promega) following the manufacturer's protocol. Lentiviral supernatants were collected after 48 h or 72 h of transfection and used to infect CH12 cells by spin-infection at 2350 RPM at 37 °C in the presence of 0.4 μg/ml polybrene (Sigma). After 24 h of infection, puromycin (Sigma) was added at a final concentration of 1 μg/ml to select for infected cells. Cells were harvested 48 h later.

## Sample preparation for replication timing (RT) analysis

Repli-seq was performed as previously described[40]. In brief, two million asynchronously dividing cells were seeded and incubated with 100 μM BrdU (Sigma) for 2 h in a light-protected environment to maintain BrdU stability. Cells were fixed and incubated with a mix of RNase A (Invitrogen) and propidium iodide (Sigma) for 30 min (light-protected). For each sample, three fractions were sorted: G1 phase, early S phase and late S phase cells, and for each fraction, two independent samples of 50,000 cells (technical replicates) were sorted on a Sony SH800S Cell Sorter. Sorted cells were lysed with Proteinase K buffer overnight. Extracted DNA was sonicated for 9 min in a Diagenode Bioruptor resulting in 100–500 bp DNA fragments as determined on an agarose gel. Sonicated DNA was subjected to end-repair and adapter ligation using the NEBNext® Ultra™ II DNA Library Prep Kit (NEB) following the NEB protocol. Adapter-ligated DNA was incubated with 25 μg/ml of anti-BrdU antibody (BD Pharmingen) for 4 h with rotation followed by incubation with 40 μg of anti-mouse IgG antibody (Sigma) for 1 h with rotation (light protected). DNA was precipitated via Centrifugation at 16,000 g for 5 min at 4 °C. Pellet was resuspended in 200 μl of digestion buffer (for 50 ml of digestion buffer, combine 44 ml of autoclaved double-distilled water, 2.5 ml of 1 M Tris-HCl pH 8.0, 1 ml of 0.5 M EDTA and 2.5 ml of 10% SDS) with freshly added 0.25 mg/ml proteinase K and incubated overnight at 37 °C. The immunoprecipitated DNA was used for Repli-qPCR or next-generation sequencing (Repli-seq). Libraries that were successfully validated by Repli-qPCR were sequenced on an Illumina HiSeq 2500 machine (50 bp, single-end). Up to 12 barcoded samples were pooled per lane.

## Preparation of chromatin fraction from CH12 cells

The chromatin faction was prepared as described[74]. In brief, cells were lysed in lysis buffer (20 mM Tris pH 7.5, 150 mM NaCl, 5 mM MgCl₂, 0.2% NP40, 5% glycerol, 1 mM DTT) supplemented with PhosSTOP phosphatase inhibitor cocktail (Roche) and cOmplete™ EDTA-free protease inhibitor cocktail (Roche). Following centrifugation, the pellet was washed twice in lysis buffer, resuspended in lysis buffer and incubated for 10 min with Benzonase (made in-house by the Molecular Biology Service, IMP, Vienna). Protein concentration was measured with the Bradford assay and proteins were analyzed on a 4-12% SDS-PAGE gel (Invitrogen). Western blot analyses were performed with MCM4 (polyclonal; ab4459 from Abcam; 1:1000 dilution) and RNA Pol II (clone 4H8; ab5408 from Abcam; 1:2000 dilution) antibodies.

## Chromatin immunoprecipitation

Chromatin immunoprecipitation was performed as described[36]. In brief, primary B cells were isolated from *Rif1*^FH/FH and C57BL/6 (*Rif1*^WT/WT) mice spleens using anti-CD43 MicroBeads (Miltenyi Biotec) and

expanded in complete RPMI containing 5 μg/ml LPS (Sigma-Aldrich) and 5 ng/ml mouse recombinant IL-4 (Sigma-Aldrich) to allow B cell activation and class switch recombination to IgG1. B cells were harvested 72 hr after activation and $4 \times 10^7$ cells were cross-linked by using 2 mM disuccinimidyl glutarate (ThermoFisher 20593) in PBS for 45 min and 1% formaldehyde for 5 min (Thermo Scientific 28908). The reaction was quenched with 0.125 M glycine. Cells were washed thrice with ice-cold PBS and lysed in SDS lysis buffer. Chromatin fragmentation was performed using a Covaris E220 sonicator to obtain fragments between 200 bp and 600 bp. Chromatin was quantified with a ND-1000 NanoDrop spectrophotometer. Immunoprecipitation was performed with 0.5 μg of anti-HA antibody (Santa Cruz sc-7392) and 50 μl of Dynabeads protein G (Thermofisher 10003D) and 25 μg chromatin. ChIP libraries were prepared by NEBNext ultra II DNA library preparation kit (NEB E7645L) and sequenced on one lane of a NovaSeq 6000 (Illumina) machine.

For ChIP-qPCR analyses at ISs, samples were prepared as above and immunoprecipitated with an MCM5 polyclonal antibody (5 μg for 5 million cells) (Bethyl Laboratories, A300-195A). A list of ChIP-qPCR primers is provided in Supplementary Data 3.

## RNA-seq from CH12 cells

RNA-seq was performed on CH12 and two independent RIF1-deficient (*Rif1*^−/−) CH12 clonal derivatives[37] with three replicates. Cells were expanded in complete RPMI and 1 million cells were collected by centrifugation. RNA was extracted with TRIzol (Invitrogen) according to manufacturer's instructions. TruSeq RNA Library Prep Kit v2 (Illumina) was used to prepare a whole-transcriptome sequencing library and sequenced on one lane of a NovaSeq 6000 SP (Illumina) machine.

## Isolation of short nascent strands (SNSs)

SNSs were isolated following an established protocol[51] kindly provided by Dr. Maria Gomez (CBMSO, Madrid) and also described in our previous study[39]. In brief, 200 million cells per replicate from asynchronous cell cultures were harvested and genomic DNA was extracted. DNA was denatured at 95 °C for 10 min and then subjected to size fractionation via 5-20% neutral sucrose gradient centrifugation (24,000 g for 20 h). Fractions were analyzed by alkaline agarose gel electrophoresis and those in the 500–2000 nt range were pooled. Prior to all following enzymatic treatments, ssDNA was heat denatured for 5 min at 95 °C. DNA was phosphorylated for 1 h at 37 °C with T4 Polynucleotide kinase (NEB or made in-house by the Molecular Biology Service, IMP). To enrich for nascent DNA strands, the phosphorylated ssDNA was digested overnight at 37 °C with Lambda exonuclease (NEB or made in-house by the Molecular Biology Service, IMP). Both T4 PNK and Lambda exonuclease steps were repeated twice for a total of three rounds of phosphorylation and digestion. After the final round of digestion, the DNA was treated with RNaseA/T1 mix (Thermo Scientific) to remove 5′ RNA primers and genomic RNA contamination. DNA was purified via phenol-chloroform extraction and ethanol precipitation This material was either used directly for qPCR or further processed for library preparation.

## SNS library preparation for SNS-seq

SNSs prepared as described above were converted to double-stranded DNA (dsDNA) via random priming with random hexamer primer phosphate (Roche) and ligation with Taq DNA ligase (NEB)[51]. DNA was checked on a fragment analyzer and 50 ng was used for library preparation with the NEBNext® Ultra™ II DNA Library Prep Kit (NEB) following the manufacturer's protocol. Libraries were barcoded using the NEBNext® Singleplex Oligos (NEB) as per the NEB protocol which allowed 4–8 libraries to be pooled per run. Sequencing was performed on an Illumina HiSeq 2500 machine (50 bp, single-end).

## In situ Hi-C

Hi-C was performed as previously described[45] with minor modifications. In brief, 5 million cells were crosslinked with 1% formaldehyde (Sigma) for 10 min and quenched with 0.6 M glycine for 5 min. Cells were lysed with Hi-C lysis buffer for 1 h on ice and nuclei were collected by centrifugation. Nuclei were digested with 375 U of MboI (NEB) overnight at 37 °C with rotation. Biotin-14-dATP (Life Technologies) was incorporated for 1 hr at 37 °C with rotation. Ligation of overhangs was performed with 20,000 U of T4 DNA ligase (NEB) for 4 h at room temperature with rotation. Nuclei were pelleted and reverse-crosslinked overnight. Purified DNA was sonicated for 14 min in a Diagenode Bioruptor to obtain a size range of 200–700 bp. This material was purified using Agencourt AMPure XP beads (Beckman Coulter). Between 8 and 15 μg of DNA was incubated with 100 μl (~10 mg) of Dynabeads MyOne Streptavidin C1 (Invitrogen) for 15 min with rotation. End repair and adapter ligation using the NEBNext® Ultra™ II DNA Library Prep Kit was performed on-beads following the kit manual. The adapter-ligated DNA was washed, eluted and PCR-amplified with KAPA 2X HiFi HotStart ReadyMix (Kapa Biosystems) and the NEBNext Multiplex Oligos for Illumina® (Dual Index Primers Set 1). Four pooled, barcoded samples were sequenced on an Illumina NovaSeq 6000 machine (50 bp, paired-end).

## PRO-seq

PRO-seq was performed as described previously[75] with minor modifications. To isolate nuclei, CH12 cells and *Drosphila* S2 cells were resuspended in cold Buffer IA (160 mM sucrose, 10 mM Tris-Cl pH 8, 3 mM CaCl$_2$, 2 mM Mg acetate, 0.5% NP-40, 1 mM DTT added fresh), incubated on ice for 3 min and centrifuged at 700 g for 5 min. The pellet was resuspended in nuclei resuspension buffer NRB (50 mM Tris-Cl pH 8, 40% glycerol, 5 mM MgCl$_2$, 0.1 mM EDTA). For each run-on, 10 million CH12 nuclei were spiked with 10% *Drosophila* S2 nuclei in a total of 100 μL NRB and incubated at 30 °C for 3 min with 100 μL 2x NRO buffer including 5 μl of each 1 mM NTP (biotinylated ATP and GTP, and unlabelled UTP and CTP). The following steps were performed as described[75] with the following changes: (1) we used different adapters, namely, 3′ RNA adapter 5Phos/NNNNNNNGAUCGUCGGACUGUA-GAACUCUGAAC/3InvdT-3′ and 5′ RNA adapter: 5′-CCUUGGCACCC-GAGAAUUCCANNNN-3′. (2) 3′ and 5′ ligations which were done at 16 °C overnight, and (3) CapClip pyrophosphatase (Cellscript) used for 5′ decapping. RNA was reverse transcribed by SuperScript III RT (Invitrogen) with RP1 Illumina primer to generate cDNA libraries. Libraries were amplified using barcoding Illumina RPI-x primers and the universal RP1 and KAPA HiFi Real-Time PCR Library Amplification Kit. Amplified libraries were subjected to electrophoresis on 2.5% low melting agarose gel and amplicons from 150 to 350 bp were extracted from the gel, multiplexed and sequenced on Illumina platform Next-Seq 550 SR75. Bioinformatics analyses were performed as described[75] but additionally the random 8-mer was used to exclude PCR duplicates and only deduplicated reads were aligned.

## Biological and technical replicates

For biological replicates, different frozen vials of CH12 cells were thawed and kept separate throughout the course of the experiment. Replicates for all next-generation sequencing experiments were derived in this manner. Technical replicates, where used (such as in RT-qPCR or ChIP-qPCR), were subsets of the biological replicate.

## Statistical analyses

For correlation scatter plots, the Pearson correlation coefficient was calculated to determine the degree of correlation. In all other cases, the two-tailed Student's t test was used for statistical significance. For RNA-seq analysis with DESeq2, a Wald test is used on the estimated log2 fold changes followed by a Benjamini-Hochberg multiple test correction.

## Bioinformatics

Main next-generation sequencing data analysis workflows for SNS-seq, Repli-seq and Hi-C have been wrapped with the *Nextflow* workflow language[76] and are centrally available at https://github.com/pavrilab[13]. The usage of Docker and Singlularity containers ensures portability, reproducibility and reliability for all workflows. Along with integration of GitHub repositories for self-contained pipelines all workflows are easily ported to all major HPC computing platforms such as SGE, SLURM, AWS. Furthermore, continuous checkpoints for pipeline execution allow for resuming and automatic retrial of failed steps. All workflows produce elaborate QC-reports and out-of-the-box, resource consumption reports to allow tailoring resource requirements to your datasets which especially for Hi-C datasets can vary by several orders of magnitudes.

## SNS-seq analysis

SNS-seq data was analysed with the *inisite-nf* pipeline. In brief, raw reads were adapter- and quality-trimmed using trim_galore v0.6.4 (https://github.com/FelixKrueger/TrimGalore) discarding reads shorter than 18 bp. Trimmed reads were aligned to the reference genome (mouse genome build UCSC mm9) with bowtie v1.2.3[77] (-v 2 –best –strata –tryhard –m 1 –chunkmbs 256). Peaks were called from the resulting alignments using MACS v2.2.6[46] (--nomodel –extsize 275 -q 0.01) without input and consensus peaks for each condition were computed from replicates using BedTools intersect v2.27.1[78]. Consensus peaks were then clustered using ClusterScan v0.2.1[79] setting -n 2 and -d to the median interpeak distance of the respective consensus peak set and subsequently filtered with BedTools intersect v2.27.1[78] to only retain peaks within a cluster. Resulting peak sets for each condition pair compared were then merged using BedTools merge v2.27.1[78].

## RNA-seq analysis

Reads were adapter trimmed using cutadapt v1.4.2[80] (--match-read-wildcards -O 1 -a AGATCGGAAGAGCACACGTCTGAACTCCAGTCAC) discarding reads shorter than 18 bp. Trimmed reads were filtered for rDNA contamination (gi|374088139, gi|38176281) with bowtie2 v2.1.0[81] (--very-sensitive-local). Non-matching reads were then aligned against the reference genome (mouse genome build UCSC mm9) with STAR v2.4.2[82] (--outSAMstrandField None --outFilterIntronMotifs RemoveNoncanonical --outFilterMismatchNoverLmax 0.1 --outFilterMismatchNmax 10 --outFilterScoreMinOverLread 0.30 --outFilterMatchNminOverLread 0.30 --outFilterMatchNmin 30 --chimSegmentMin 15 --quantMode TranscriptomeSAM --chimJunctionOverhangMin 15 --twopassMode Basic --outSAMtype SAM --outSAMattributes All --outReadsUnmapped Fastx intronMotif --alignIntronMax 200000 --outSJfilterIntronMaxVsReadN 10000 20000 30000 50000 --outSJfilterOverhangMin 20 12 12 12 --outFilterType BySJout --alignMatesGapMax 0 --outFilterMultimapNmax 20) using an index built from the mm9 reference genome and a gene annotation created from all refGenes and ensGenes tables downloaded from the UCSC table browser on March 1$^{st}$ 2014. Read counts were quantified using featureCounts v2.0.0[83] with the mm9 refGenes annotation downloaded from the UCSC table browser on February 28$^{th}$ 2020 together with *Igh*, *Igk* and *Igl* genes (extracted from the Gencode M25 annotation using Ensemble Gene IDs from IMGT; http://www.imgt.org, August 11, 2020). *Igh*, *Igk* and *Igl* genes were lifted over to mm9 coordinates and added to Elambda 3_1 enhancer[84]. Differential gene expression analysis was performed with DESeq2 v1.22.2[44] with log2 fold change shrinkage using the ashr package v2.2-47[85] employing a threshold-based Wald test for assessing significance with a log2 fold change threshold of <1.

## Repli-seq analysis

Raw Repli-seq data was processed using the *repliseq-nf* pipeline which loosely follows the steps describe in[40]. In brief, raw reads of early (E) and late (L) fractions were adapter and quality trimmed with

trim_galore v0.6.5 (https://github.com/FelixKrueger/TrimGalore). Trimmed reads were aligned to the mouse mm9 reference genome with bwa mem v0.7.17[86] and resulting alignments were filtered for unique alignments with samtools v1.9[87]. Replicates were merged with samtools v1.9[87] and duplicated reads were removed using picardTools MarkDuplicates (https://broadinstitute.github.io/picard/). E/L log2 ratios were calculated in 20 kb windows using deepTools bamCompare v3.4.1[88] and subsequently loess smoothed with a custom R script using the preprocessCore package v1.46.0 (https://github.com/bmbolstad/preprocessCore) with a span size of 300 kb. BigWig tracks were produced from this using kent_tools v377[89].

## Hi-C analysis

Raw Hi-C data was processed with the *hicer-nf* pipeline v1.0.0. In brief, raw Hi-C reads were adapter and quality trimmed with trim_galore v0.6.5[90] (https://github.com/FelixKrueger/TrimGalore) and aligned and filtered for data-specific artefacts using the HICUP pipeline v0.7.3[90] together with bowtie2 v2.3.5.1[81]. Resulting alignments were then used to produce contact matrices in COOL format using cooler csort, cload and zoomify v0.8.6[91] which were subsequently balanced with the C++ implementation of the KR algorithm from HiCExplorer[92]. Eigenvectors for AB compartmentalization were computed with HOMER v4.10[93] using a binsize of 20 kb and smoothing window of 200 kb. Sign correctness of the eigenvectors values was assessed by correlation with the gene annotation. Saddle plots were computed from the 20 kb eigenvectors with cooltools v0.3.2 binning the eigenvectors into 50 equal-sized bins.

## Generating heatmaps for IS peak summit neighborhoods

Sequencing coverage normalized bigWig tracks were generated from raw aligned reads of NGS data sets with deeptools bamCoverage v3.3.0[88] using command-line parameters --normalizeUsing CPM --exactScaling and --ignoreDuplicates. Next, we computed the position of the replication start site (RSS) as the genomic coordinate of the maximum pile-up of the merged tracks of compared conditions. The generated summit positions were then grouped by their associated peak class and sorted in decending order on the log2-ratio of CPM values of treatment versus control. The sorted summits were then used as reference points for deeptools computeMatrix v3.3.0[88] to compute signal distributions for previously generated CPM-normalized bigWig tracks within a 4 kb region centered on the summit using a binsize of 50 bp and setting missing values to zero. The results were then plotted using the associated plotHeatmap command.

## PRO-seq analysis

The 3′ end sequence of the reads (NNNNTGGAATTCTCGGTGCC) was removed using cutadapt v1.4.2[80] and 9 nucleotides from their 5′ ends were removed to remove the random 8mer and in vitro run-on nucleotide. The trimmed reads were reverse complemented using fastx_reverse_complement (http://hannonlab.cshl.edu/fastx_toolkit; version 0.0.13) followed by deduplication based on the 8mer sequence. Trimmed reads longer than 18 bp were aligned to a hybrid mouse drosophila genome (mouse genome build NCBI m37, GCA_000001635.18, drosophila FlyBase release 5) using bowtie v1.0.0[77] with -v 2 --best --strata --tryhard -m 1 --chunkmbs 256. Unique mappers from the resulting BAM file were used to create bigwigs with deeptools bamCoverage 3.3.0.

## RT domain analysis

20 kb binned RT tracks for all conditions were segmented into three states using hmm_bigwigs (https://github.com/gspracklin/hmm_bigwigs). Fitted states were then mapped to either early (E), mid (M) or late (L) replication timing based on the RT value distribution of each state (E for RT > 0, L for RT < 0 and M for RT ~ 0). M state bins were split at 0 into L-like (values < 0) and E-like (values > 0) state bins.

## Generating saddle plots from Hi-C data

Saddle plots were computed using cooltools v0.3.2. In brief, we subdivided all bins of a 20 kb KR-normalized contact matrix into 50 equal-sized groups based on the bins compartment signal as derived from the eigenvector of the WT data, where group 1 has the lowest signal (i.e. most B) and group 50 has the highest signal (i.e. most A). Subsequently, we compute the mean observed/expected value for each pair of groups and plot it as a 50 × 50 matrix. Similarly, replication timing and H3K9me3 data can be used for bin group assignment.

## RIF1 peak and RIF1 associated domains (RADs) analyses

Peaks and domains (RADs) were called as described previously[36] additionally calling narrow peaks with MACS2. In brief, raw ChIP-seq and input reads in MEFs and primary B cells were trimmed with trim_galore v0.6.4 (https://github.com/FelixKrueger/TrimGalore) setting minimum length to 18 bases and default arguments otherwise and aligned to the mm9 reference genome with bowtie v1.0.0[77] with -S --trim5 0 --trim3 0 -v 2 --best --strata --tryhard -m 1 --phred33-quals --chunkmbs 256. Subsequently, RIF1 narrow and broad peaks were called for each replicate with MACS v2.2.6[46], RIF1 associated domains (RADs) were called using EDD v1.1.19[47]. RADs, narrow and broad peaks were merged using bedtools v2.27.1[78] and only those peaks called in both replicates were retained for downstream analysis. Genome coverage tracks were computed with deeptools bamCoverage v3.3.0[88] using command-line parameters --normalizeUsing CPM --exactScaling --binSize 10 and --ignoreDuplicates. Log2 ratio tracks were computed with deeptools bamCompare v3.3.0[88] using command-line parameters --scaleFactorsMethod readCount --operation log2 --pseudocount 1 --binSize 50 --ignoreDuplicates.

## Generating clustered heatmaps of RIF1 binding sites

RIF1 ChIP-seq was used together with datasets measuring nascent transcription (PRO-seq for primary B cells[94] and GRO-seq for MEFs[50]), chromatin accessibility (ATAC-seq[95]) and levels of histone modifications (H3K27Ac and H3K9me3[96] ChIP-seq) to compute heatmap data within a 4 kb region centered on the middle of RIF1 peaks using deeptools computeMatrix v3.3.0[88] with a bin size of 50 bp and setting missing values to zero. RIF1 peaks were then grouped based on nascent transcription, ATAC-seq and ChIP-seq data using k-means clustering with k = 3. The resulting clusters were sorted based on RIF1 ChIP-seq heatmap signal intensity and subsequently plotted. Clustering, sorting and plotting was done using deeptools plotHeatmap[88].

## Summary of all next-generation sequencing data and analysis software

All data sets used in this study are listed in Supplementary Data 4. The software used for analysis is provided in Supplementary Data 5 and divided into three tabs (general, r and python packages).

## Reporting summary

Further information on research design is available in the Nature Portfolio Reporting Summary linked to this article.

## Data availability

The next generation sequencing data (Repli-seq, RNA-seq, Hi-C, ChIP-seq and SNS-seq) generated in this study have been deposited in NCBI's Gene Expression Omnibus and are accessible through GEO Series accession code GSE228880. Source data are provided with this paper.

## Code availability

The data processing workflows for the next-generation sequencing datasets with data analysis scripts are available at https://github.com/pavrilab with stable version releases provided on Zenodo as follows: Repli-seq[97], Hi-C[98], SNS-seq[99] and all other analyses[100].

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

## Acknowledgements

We are grateful to the Vienna Biocenter Core Facilities (VBCF) for next-generation sequencing and the IMP/IMBA core facilities especially the animal house, bio-optics, molecular biology service and proteomics service. We are grateful to Roman Stocsits and Dr. Anton Goloborodko for advice on Hi-C data analysis. This work was funded by Boehringer Ingelheim, The Austrian Industrial Research Promotion Agency (Headquarter Grant FFG-834223), grants from the Austrian Science Fund to RP (FWF P 29163-B26) and UES (FWF T 795-B30), and the Helmholtz-Gemeinschaft Zukunftsthema "Immunology and Inflammation" ZT810 0027 to MDV and AR.

## Author contributions

D.M. performed most of the bioinformatics analyses, established pipelines and analyzed data. M.P. performed Repli-seq, Hi-C, RNA-seq and SNS-seq, and analyzed data. A.R. performed RIF1 ChIP-seq from primary B cells and RNA-Seq in CH12 cells. S.G. and K.N.K. performed Repli-seq from primary B cells. R.P. performed MCM4 phosphorylation analysis and MCM5 ChIP-qPCR. U.S. performed PRO-seq. M.N. collaborated on Repli-seq from CH12 cells. D.M.G., S.C.B.B. and M.D.V. contributed personnel, resources, extensive discussions and critical reading of the manuscript. T.N. and R.P. co-supervised the project and analyzed data. R.P. conceived the project and wrote the manuscript with inputs from all authors.

## Competing interests

The authors declare no competing interests.
