## [Peer Review File · Nature Communications]

RIF1 regulates early replication timing in murine B cellsREVIEWER COMMENTS

Reviewer #1 (Remarks to the Author):

Regulation replication timing is a critical for appropriate cell division and preservation of genomic integrity. The mechanisms that coordination of early versus late origin firing remain poorly understood. In this manuscript, the authors investigate the function of RIF1 in regulation of replication timing in activate B cells. They find that RIF1 controls early replicating regions and loss of RIF1 disrupts timing of these early replication sites to a greater extent that late replicating regions. This is a new and unexpected function of RIF1, which has been previously demonstrated to suppress late origins of replication in heterochromatin. The manuscript is well-written with well-controlled and complementary experiments. The authors do a very nice job distilling complex experiments and results into understandable and interpretable findings for readers. The results are novel and establish new insights into replication control in B cells as well as provide foundation for cell type-specific mechanisms for replication timing control.

A few minor comments:

1. In Figures 1 through 5, the green (Rif1^{-/-}) and blue (WT) colors are difficult to distinguish in some of the figures, particularly in the UCSC browser tracings. More distinct color options would make it easier for readers to interpret results.

2. Line 183 on page 7, the text should reference Fig. 2G (rather than Fig. 2F).

3. Line 299 on page 9, text should reference Fig. 4A (rather than Fig. 4B).

4. Line 242-246, text states "The results showed that in WT cells, class 1 ISs were the most active (Fig. 4C)...whereas class 5 ISs were least active...(Fig. 4C-D)." It's not evident that Fig. 4C provides results on which ISs are most versus least active. The graphic under Fig. 4C-D delineates activity of ISs in WT versus Rif1^{-/-} but the data in Fig 4C shows density of ISs in each class, which is not indicative of activity. Can the authors clarify this in the text?

5. In Figures 5-7, MCM6 has the dominant role in regulation of replication timing compared to RIF1. Loss of MCM6 has more significant impact on replication timing in all the assays. Loss of RIF1 in the MCM6-deficient cells does lead to additional changes supporting that the two proteins are complementary and non-epistatic. However, it should be stated/presented in these result sections that MCM6 has the dominant phenotype. The authors do address this in the Discussion but this information should also be included in the presentation and interpretation of the data in the Results.

Reviewer #2 (Remarks to the Author):

This manuscript investigates the role of Rif1 in activated B cells. The results presented are very unexpected, and potentially present a previously unknown function of Rif1.

Rif1 has been known to suppress the firing of replication origins by binding to chromatin in the vicinity of late-firing origins or in the late replicating regions. In this manuscript, authors show that Rif1 binds to early replicating domains and promotes early replication. The data supporting this conclusion appears to be robust and convincing. The data analyses have been conducted in a professional manner.

Major comments:

1 Looking at the data in Figure 1B, it appears that loss of Rif1 induces both E to L and L to E

conversions, resulting in RT with less dynamic range (toward flat RT), especially in primary B cells. This is similar to what has been observed in human iPS cells.

Does Rif1 binds selectively to early replicating domains in this chromosome region (chr2: 57,676,011-121,972,628)? It is known, in some cancer cells, that Rif1 binds to selective segments of chromosomes and RT of not all the chromosomes are regulated by Rif1. Do all the early replicating domains respond to Rif1 depletion by causing later replication? Or is there any segmental selectivity in RT regulation by Rif1 in activated B cells?

I see the transition of the late replicating domains to earlier-replicating in Rif1 $-/-$ cells (Figure 1D). How is the Rif1 binding profile in those segments, where early replication is not affected or where late replication is converted to early by loss of Rif1?

2 In fission yeast Rif1 mutant, E to L transmission is observed at a number of early replicating regions (Hayano et al. 2012; Genes and Dev), and it has been explained that rate-limiting replication factors are utilized at activated late origins, and cannot be used for early origins, thus reducing origin firing at some early firing origins. A possibility remains that, due to activation of late origins which utilizes initiation factors, early origins cannot be fully activated due to the potential low abundance of initiation factors in activated B cells. This possibility needs to be discussed.

3 Authors state as follows (line 206). "Peaks in both cell types were enriched in early-replicating regions (Fig. 3C-D). However, although RADs in MEFs were mostly late-replicating, as described, in B cells, they were largely early-replicating (Fig. 3C-D)."

It is not intuitively clear that RADs (RIF1-associated domains) are late replicating, while Rif1 binding peaks are in early-replicating regions in MEFs. How are the peaks are enriched in the early-replicating regions, but Rif1 associates mainly in the late replicating regions?

Some more explanation is needed here for lay people to this field to be able to understand the points.

4 I suggest authors conduct several experiments described below to improve the manuscript before it is further considered for publication in NC.

4-1 Mechanistic insight into how Rif1 may directly regulate early firing origins is lacking in this manuscript.

It has been well established that Rif1's ability to recruit PP1 plays a major role in its ability to suppress origin firing. Authors need to examine the PP1 binding mutant of Rif1 in B cells to determine whether PP1 recruitment is required for early origin promotion in B cells. Authors also need to examine the phosphorylation states of MCM4 and Treslin, which are known to be enhanced in Rif1-depleted cells.

4-2 Figure 3E

The data in MEF needs to be shown in comparison.

H3K9 methyl pattern also needs to be shown.

It has been reported that Rif1 binding sites and H3K9me3 do not overlap.

4-3 Authors propose that Rif1 loss leads to reduced licensing, causing less efficient early origin firing. This possibility is testable by Mcm ChIP-seq, along with the data showing the extent of licensing in Mcm6 knockdown cells. This will also test the authors hypothesis that licensing is lost specifically in highly transcribed regions in the absence of Rif1.

4-4 Figure 5

The extent of Mcm6 depletion in both Rif1+ and Rif1 $-/-$ cells need to be shown by western analyses.

Minor comments:

It would be helpful for readers if authors clearly state, in each figure, which cells were used for analyses, primary B cells or CH12 cells.

Line 181; Fig2.E-F should be Fig2.E-G, Line 183; Fig2.F should be Fig2.F-G, Line 191; Fig2.E-F should be Fig2.E-G,

Figure 1B, 2C, 5D, 6D
sMcm6 -> shMcm6

Fig.5C (lower panel)

It appears that Rif1^{-/-} advances RT of mid-origins, consistent with other cell types. Authors need to discuss this.

Fig. 1G

Y axis needs to be explained in the legend.

Fig. 3E

The correlation between Pro-seq peaks and RIF1-ChIP-seq peaks appears not to be very strong. What is the extent of overlap between Pro-seq and Rif1-ChIP peaks?

Reviewer #3 (Remarks to the Author):

This manuscript describes the effect of Rif1 on DNA replication in murine B lymphocytes, and reports the surprising finding that Rif1 associates most strongly with early-replicating chromosome domains, and that removal of Rif1 delays the replication of these normally early regions. Numerous previous studies have shown that Rif1 generally specifies late replication in both mouse and human cells, at least partly by directing Protein Phosphatase 1 to dephosphorylate the MCM replicative helicase and possibly other targets. In this study, the authors used the Repli-seq method to examine replication timing (RT) in Rif1^{-/-} mouse CH12 cells and in mouse primary B cells. In contrast to previous findings in other cell lines and to the established model of its action, they find that Rif1 is important for specifying the replication time of the early replicating regions. In some other cell lines Rif1 has been implicated by 3C analyses in specifying chromosome architecture, but in mouse B cells any effect on genome architecture is minor. In a particularly interesting and strong experiment, in Fig. 3 the authors use Repli-seq and ChIP-seq to show that in B cells Rif1 is specifically bound to early domains, in complete contrast to the situation in MEFs where Rif1 associates with late domains as previously reported in the literature. This surprising result certainly gives an entirely new view of how Rif1 may affect DNA replication. However, the authors' proposal that Rif1 directly regulates early origins is weaker: SNS-seq is a controversial method and even accepting the data shown in Fig. 4 at face value, only a few of the very strongest origins are affected. Nor is any data presented indicating that Rif1 interacts specifically or directly with origin sites. Still, the interpretation of this SNS-seq experiment does not affect the main conclusions of the paper concerning Rif1 association domains and effects on timing. The authors go on to show that MCM and Rif1 act additively in B cells to affect early RT and early origin activity, with MCM having a noticeably stronger effect. Perhaps surprisingly, even depleting both MCM and Rif1 does not affect chromatin organisation. Finally, the authors make the interesting observation that in Rif1^{-/-} or shMcm6 B cells the regions whose replication timing is most compromised are those that show the highest levels of transcription. Based on this last finding the authors provide an interesting Discussion suggesting reasons why Rif1 might have been re-deployed in B cells to have this special effect on replication, and relate their findings to an earlier study where human Rif1 was shown to stabilise Orc1 on chromatin to promote origin licensing. Overall, given the interesting new findings the paper is a good candidate for Nature Communications. The following points should be addressed:

Major Points

1. In places the textual explanations are insufficient and/or opaque for the non-expert. In several

places more detailed explanation should be provided explaining the aspects on which the reader should focus (e.g., for Fig. 1C, Fig. 2E and Fig. 3E).

2. The Title and Abstract should specify that the study examines mouse cells.
3. Although it's clear that the major effect of Rif1 KO is on early regions, there is also some advancement on the replication time of late regions, seen for example in Fig. 1B and 1D. This effect, consistent with previous studies of Rif1, is overly downplayed in the main text.
4. Saddle plots in Fig. 2F-G seem to over-represent/exaggerate the changes in chromatin contacts that are seen in Fig. 2E. Therefore, Fig. 2F-G could be omitted.
5. The co-localisation of Rif1 peak intensity with transcription, chromatin accessibility and the H3K27ac mark is mild/weak in each case in Fig. 3E. This should be mentioned in the text.
6. Fig. 4B uses SNS-seq to identify sites of initiation, which is controversial as the technique may assign some sites of re-priming as origins of replication. Given this issue, the mildness of the effect in Fig. 4B, and the fact that Rif1 does not bind origin sites directly but rather early domains, the authors should alter their abstract to remove the words 'regulates early origin firing', and moderate their claims that Rif1 directly controls origin initiation.
7. The meaning of the vertical 'lines' observed through the centre of the DHS-seq and H3K36me3 experimental datasets in Fig. 4B is unclear. Could the authors elaborate on the reason for these low signal zones?
8. It would be helpful to move Fig. S4G into Fig. 5 to show the density-contour plots of Rif1^{-/-} and Rif1^{-/-} shMcm6 cells together with their effects on RT and chromatin architecture (using Hi-C).
9. The scatter plot in Fig. 6A regarding Rif1^{-/-} shMcm6 cells is not accompanied by the same heatmaps as in Fig. 4B for Rif1^{-/-} cells. This data should be included in Fig. 6.
10. Could the authors test whether the positive effect of Rif1 on early origin firing is dependent on PP1? This would further strengthen their model as outlined in the discussion section.

Minor Points

1. Often 'shMcm6' labelling on figures is mislabelled as 'sMcm6'. (e.g., Fig. 1B and Fig. 2C).
2. An experimental condition label (i.e., shMcm6 vs. shLacZ) on top of each panel would make it clearer what each experiment is showing (e.g., Fig. 1C).
3. Fig. S1A and S1C and Fig. S3A-B show the reproducibility of each Repli-seq, SNS-seq or ChIP-seq experiment, but are not particularly informative for the general reader and could be less prominent.
4. The authors could associate an 'Early' or 'Late' label with the positive or negative values in each figure where RT is examined (e.g., Fig. 1B). At least for this Reviewer, the graphs could be read more intuitively with the x-axis reversed so 'Early' is on the left and 'Late' on the right.
5. The blue and red colours used in Fig. 1F and 1G are confusing as they don't mean the same thing (i.e., one denotes RT and the other denotes an experimental condition e.g., shMcm6).
6. In Fig. 2C, given the fairly minor changes in RT that occur in Rif1^{-/-} conditions (i.e. Early regions somewhat delayed, but still Early) we wouldn't expect much of a change in PC1 values. So, the conclusion drawn is perhaps not as surprising as the authors suggest and is somewhat overstated.
7. The lane order between Fig. 3C and Fig. 3D is changed, which is confusing when trying to compare RT and Rif1 localisation in the primary B and MEF cell lines.
8. The inclusion of the word 'decreasing' in the y-axis label of Fig. 3E is confusing. The authors should omit this word but keep the grey bar indicating peak intensity.
9. A couple of figures are mislabelled in the main text (e.g., Fig. 2F on page 7 should read Fig. 2G). Fig. 4A is not referred to in the main text.
10. The data in Fig. 4B is presented with the 'strongest' origins at the bottom, which is confusing, please invert these pile-up plots.
11. The black colour used in Fig. 6D to denote overlaps should be changed to be more translucent as it masks the data underneath it.
12. Fig. 7A can be omitted as Fig. 7B makes the point in a clearer and more intuitive manner.
13. The sentence in lines 385-388 is confusing and should be rewritten to clarify (do the authors mean to say 'earlier' rather than 'later' in line 388?)

**Response to reviewers**

We would like to thank the reviewers for their time, appreciation of our work and constructive
comments on our manuscript. We are happy that all reviewers appreciated our new findings on the
role of RIF1 in early RT in B cells. We have incorporated many of the suggestions from the reviewers
which have considerably strengthened the manuscript. Please find below our point-by-point response
to all comments in blue text. All new text in the main manuscript is also in blue text.

**Reviewer #1 (Remarks to the Author):**

Regulation replication timing is a critical for appropriate cell division and preservation of genomic
integrity. The mechanisms that coordination of early versus late origin firing remain poorly understood.
In this manuscript, the authors investigate the function of RIF1 in regulation of replication timing in
activate B cells. They find that RIF1 controls early replicating regions and loss of RIF1 disrupts timing
of these early replication sites to a greater extent that late replicating regions. This is a new and
unexpected function of RIF1, which has been previously demonstrated to suppress late origins of
replication in heterochromatin. The manuscript is well-written with well-controlled and complementary
experiments. The authors do a very nice job distilling complex experiments and results into
understandable and interpretable findings for readers. The results are novel and establish new
insights into replication control in B cells as well as provide foundation for cell type-specific
mechanisms for replication timing control.

A few minor comments:

1. In Figures 1 through 5, the green (Rif1^{-/-}) and blue (WT) colors are difficult to distinguish in some of
the figures, particularly in the UCSC browser tracings. More distinct color options would make it easier
for readers to interpret results.

We have darkened the Rif1 track (from lighter to darker green) to make them more discernible
from the blue WT and shLacZ tracks.

2. Line 183 on page 7, the text should reference Fig. 2G (rather than Fig. 2F).

Thank you, we have done it.

3. Line 299 on page 9, text should reference Fig. 4A (rather than Fig. 4B)

Thank you, we have changed it.

4. Line 242-246, text states “The results showed that in WT cells, class 1 ISs were the most active
(Fig. 4C)...whereas class 5 ISs were least active....(Fig. 4C-D).” It’s not evident that Fig. 4C provides
results on which ISs are most versus least active. The graphic under Fig. 4C-D delineates activity of
ISs in WT versus Rif1^{-/-} but the data in Fig 4C shows density of ISs in each class, which is not
indicative of activity. Can the authors clarify this in the text?

We apologize for the lack of clarity. By “density”, we meant SNS-seq-derived read density (RPM)
of the ISs, not the density of ISs. We have changed the axis label to “ISs (RPM)”. The graphic
under the figure shows the five classes based on fold-change of IS RPMs. Hence, class 1 ISs
were most active in WT cells whereas class 5 were the least active in WT cells. We have changed
the X axis to “Fold-change in IS read density (log₂ WT/Rif1^{-/-})”. We hope this is clear. We have
also made the changes in the figure legends.

5. In Figures 5-7, MCM6 has the dominant role in regulation of replication timing compared to RIF1. Loss of MCM6 has more significant impact on replication timing in all the assays. Loss of RIF1 in the MCM6-deficient cells does lead to additional changes supporting that the two proteins are complementary and non-epistatic. However, it should be stated/presented in these result sections that MCM6 has the dominant phenotype. The authors do address this in the Discussion but this information should also be included in the presentation and interpretation of the data in the Results.

We have made this conclusion more explicit in the results section.

Reviewer #2 (Remarks to the Author):

This manuscript investigates the role of Rif1 in activated B cells. The results presented are very unexpected, and potentially present a previously unknown function of Rif1.

Rif1 has been known to suppress the firing of replication origins by binding to chromatin in the vicinity of late-firing origins or in the late replicating regions. In this manuscript, authors show that Rif1 binds to early replicating domains and promotes early replication. The data supporting this conclusion appears to be robust and convincing. The data analyses have been conducted in a professional manner.

Major comments:

1 Looking at the data in Figure 1B, it appears that loss of Rif1 induces both E to L and L to E conversions, resulting in RT with less dynamic range (toward flat RT), especially in primary B cells. This is similar to what has been observed in human iPS cells.

Does Rif1 binds selectively to early replicating domains in this chromosome region (chr2: 57,676,011-121,972,628)? It is known, in some cancer cells, that Rif1 binds to selective segments of chromosomes and RT of not all the chromosomes are regulated by Rif1. Do all the early replicating domains respond to Rif1 depletion by causing later replication? Or is there any segmental selectivity in RT regulation by Rif1 in activated B cells? I see the transition of the late replicating domains to earlier-replicating in Rif1 $-/-$ cells (Figure 1D). How is the Rif1 binding profile in those segments, where early replication is not affected or where late replication is converted to early by loss of Rif1?

In B cells, RIF1 is present on all chromosomes indicating that there is no differential binding between chromosomes. Please see the new analysis showing this (Fig. S3B-C).

Based on the scatter plots in Fig 1C for CH12 cells, most of the early bins show delayed replication (note the red stripe consisting of early-replicating bins in Fig. 1C). The results are much more striking in primary B cells (scatter plot in Fig. 1E) where all early-replicating bins show lower RT values in *Rif1* $^{-/-}$ cells, implying that all early-replicating bins show later replication in the absence of RIF1. In other words, there are almost no bins where early replication is not affected. This is also apparent in Fig. 1I.

For the *Rif1* $^{-/-}$ primary B cells, there is indeed E to L and L to E, but in the heterozygous condition, the E to L is very dominant (see also Fig. 1D). We infer from this that the E to L changes are the direct effect of decreased RIF1 (an interpretation supported by the RIF1 ChIP-seq results) and the L to E may be due to the indirect effect of having fewer early-firing origins in the full knockout that leads to advanced firing of normally late origins. Accordingly, the size of the E to L shift in *Rif1* $^{-/-}$ is stronger than the L to E shift (Fig. 1D and 1G).

Regarding RIF1 binding in late regions, our HMM analysis showed that L bins advanced their RT but did not acquire positive RT values, but some L-like bins gained positive RT values (Fig. 1E).

When we looked at RIF1 binding in these bins, only ~10% of RIF1 sites were found in L-like bins
 and even fewer in L bins (Fig. 3D).

2 In fission yeast *Rif1* mutant, E to L transmission is observed at a number of early replicating regions
 (Hayano et al. 2012; Genes and Dev), and it has been explained that rate-limiting replication factors
 are utilized at activated late origins, and cannot be used for early origins, thus reducing origin firing at
 some early firing origins. A possibility remains that, due to activation of late origins which utilizes
 initiation factors, early origins cannot be fully activated due to the potential low abundance of initiation
 factors in activated B cells. This possibility needs to be discussed.

Thank you. This is an important point and we have mentioned it in the revised Discussion (page
 15).

3 Authors state as follows (line 206). "Peaks in both cell types were enriched in early-replicating
 regions (Fig. 3C-D). However, although RADs in MEFs were mostly late-replicating, as described, in B
 cells, they were largely early-replicating (Fig. 3C-D)." It is not intuitively clear that RADs (RIF1-
 associated domains) are late replicating, while *Rif1* binding peaks are in early-replicating regions in
 MEFs. How are the peaks are enriched in the early-replicating regions, but *Rif1* associates mainly in
 the late replicating regions?

Some more explanation is needed here for lay people to this field to be able to understand the points.

We apologize for this lack of clarity, and we have explained it in the revised version (page 8).
 Peaks reflect relatively stronger sites of RIF1 occupancy whereas RADs are larger domains where
 RIF1 is more broadly distributed or where RIF1 peaks are clustered. Both peaks and RADs
 constitute bona fide regions of RIF1 binding and are not defined based on location or RT, but
 strictly using different bioinformatics tools for significant enrichment of signal over background.
 Thus, although RIF1 is detected mostly as peaks in B cells (16,043 peaks and 289 RADs), in
 principle, peaks or RADs could be located anywhere in the genome.

4 I suggest authors conduct several experiments described below to improve the manuscript before it
 is further considered for publication in NC.

4-1 Mechanistic insight into how *Rif1* may directly regulate early firing origins is lacking in this
 manuscript.

It has been well established that *Rif1*'s ability to recruit PP1 plays a major role in its ability to suppress
 origin firing. Authors need to examine the PP1 binding mutant of *Rif1* in B cells to determine whether
 PP1 recruitment is required for early origin promotion in B cells. Authors also need to examine the
 phosphorylation states of MCM4 and Treslin, which are known to be enhanced in *Rif1*-depleted cells.

We fully appreciate that the mechanism of RIF1 function in B cell RT is not addressed in this
 study. As we have now discussed more clearly in the paper, the RIF1-PP1-mediated
 dephosphorylation of MCM4 found in other cells, which is an inhibitory function as it suppresses
 origin firing, cannot explain the RIF1-dependent firing of some early origins in B cells, which is an
 activating function. It is more likely that the role of RIF1 at early origins, if PP1-dependent at all, is
 to facilitate the firing of origins, perhaps akin to the PP2A and PP4-mediated dephosphorylation of
 replication firing factors reported recently (Jenkinson et al. Mol. Cell 2023). However, we are
 unaware of any positive function of PP1 in origin firing, and therefore, we do not have any
 candidate protein whose dephosphorylation status we could test in *Rif1*^{-/-} B cells.

However, addressing the potential role of RIF1-PP1 interaction in the regulation of early origin
 firing is technically extremely complex and, even if successful, would take over a year of work.
 RIF1 function is dosage-dependent, and therefore we cannot simply transfect a plasmid encoding
 for *Rif1*-DPP1 (a RIF mutant deficient in PP1 binding) to rescue the *Rif1*^{-/-} B cells. In the paper

from our co-author, Sara Buonomo (Gnan et al. Nature Comm. 2021), the authors had to: (1)
 construct ES cells that carried one HA-tagged knock-in allele (*Rif1*-FH) and one floxed conditional
 *Rif1* allele, and then target a mini-gene in the *Rif1*-FH allele to generate the DPP1 allele.
 Moreover, they showed that *Rif1* hemizygotes already harbored some of the knock-out
 phenotypes. In summary, to build a system so complex when we do not have candidate substrates
 of PP1 to examine seems to us to go beyond the scope of this work.

 Regarding Mcm4 phosphorylation, as found by our co-author, Sara Buonomo, there are no reliable
 antibodies to detect phospho-Mcm4 in mouse cells. Importantly, however, please note that based
 on the explanation above, it is unclear how a suppressive mechanism like the dephosphorylation
 of Mcm4, even if present, would help to explain the positive regulation of origin firing by RIF1 in B
 cells. Thus, we propose that the mechanism by which RIF1 regulates early RT is unlikely to be
 linked to Mcm4-dephosphorylation. However, as mentioned above, it could still work via PP1-
 mediated dephosphorylation of other replication proteins. We have mentioned this in the revised
 Discussion.

 4-2 Figure 3E

The data in MEF needs to be shown in comparison.

 We have shown this in the revised version (new Fig. 3F). Thank you for pointing it out.

 H3K9 methyl pattern also needs to be shown. It has been reported that *Rif1* binding sites and
 H3K9me3 do not overlap.

 We have done the comparison with H3K9me3 ChIP-seq as requested (Fig. 3E-F).

 4-3 Authors propose that *Rif1* loss leads to reduced licensing, causing less efficient early origin firing.
 This possibility is testable by Mcm ChIP-seq, along with the data showing the extent of licensing in
 Mcm6 knockdown cells. This will also test the authors hypothesis that licensing is lost specifically in
 highly transcribed regions in the absence of *Rif1*.

 This is a good suggestion, but unfortunately, it is technically unfeasible. We have tried multiple
 189 times with several antibodies, but Mcm ChIP-seq has never worked for us. We are aware that this
 is a problem in the field and that many labs, including our co-authors, have struggled with it. We
 believe that the problem, in large part, may be due to the underlying biology rather than the
 antibodies. The current understanding in the field is that MCMs are loaded evenly throughout the
 genome and can subsequently slide along the DNA without ATP. In this case, different cells have
 different locations of MCMs at any given point of time, which would average out the signal in
 population-based analyses like ChIP-seq resulting in poor enrichments over background.

 4-4 Figure 5

The extent of Mcm6 depletion in both *Rif1*⁺ and *Rif1*^{-/-} cells need to be shown by western analyses.

 Please note that this was already shown in Fig. S1B and Fig. S4A.

 Minor comments:

 It would be helpful for readers if authors clearly state, in each figure, which cells were used for
 analyses, primary B cells or CH12 cells.

 We have now stated this in the figure legends for all figures,

Line 181; Fig2.E-F should be Fig2.E-G, Line 183; Fig2.F should be Fig2.F-G, Line 191; Fig2.E-F
 should be Fig2.E-G,

Thank you for pointing it out, we have made the changes.

Figure 1B, 2C, 5D, 6D sMcm6 -> shMcm6

Thank you for pointing it out, we have corrected it.

Fig.5C (lower panel)

It appears that Rif1^{-/-} advances RT of mid-origins, consistent with other cell types. Authors need to
 discuss this.

In this figure, we are comparing Mcm6 depletion in WT cells (shMcm6) to Mcm6 depletion in *Rif1*^{-/-}
 cells (*Rif1*^{-/-} shMcm6). Hence, this analysis cannot be used to infer effects of Rif1. Rather, the
 advance is most likely the result of the exacerbation of the RT phenotype in *Rif1*^{-/-} shMcm6 cells
 compared to shMcm6 alone and is reflected in the sign flips in RT values seen in Fig 5D. Please
 note that in Fig 1C, however, where we compare control and *Rif1*^{-/-} cells, there is no major
 advance of mid-origins.

Lower panel is Fig. 1G Y axis needs to be explained in the legend.

We have done so, thank you.

Fig. 3E, The correlation between Pro-seq peaks and RIF1-ChIP-seq peaks appears not to be very
 strong. What is the extent of overlap between Pro-seq and Rif1-ChIP peaks?

To address this point, also raised by another reviewer, we have provided a more detailed analysis
 of the RIF1 peaks in Fig. 3E-F and Fig. S3E.

**Reviewer #3 (Remarks to the Author):**

This manuscript describes the effect of Rif1 on DNA replication in murine B lymphocytes, and reports
 the surprising finding that Rif1 associates most strongly with early-replicating chromosome domains,
 and that removal of Rif1 delays the replication of these normally early regions.

Numerous previous studies have shown that Rif1 generally specifies late replication in both mouse
 and human cells, at least partly by directing Protein Phosphatase 1 to dephosphorylate the MCM
 replicative helicase and possibly other targets. In this study, the authors used the Repli-seq method to
 examine replication timing (RT) in Rif1^{-/-} mouse CH12 cells and in mouse primary B cells. In contrast
 to previous findings in other cell lines and to the established model of its action, they find that Rif1 is
 important for specifying the replication time of the early replicating regions. In some other cell lines
 Rif1 has been implicated by 3C analyses in specifying chromosome architecture, but in mouse B cells
 any effect on genome architecture is minor. In a particularly interesting and strong experiment, in Fig.
 3 the authors use Repli-seq and ChIP-seq to show that in B cells Rif1 is specifically bound to early
 domains, in complete contrast to the situation in MEFs where Rif1 associates with late domains as
 previously reported in the literature. This surprising result certainly gives an entirely new view of how
 Rif1 may affect DNA replication.

However, the authors' proposal that Rif1 directly regulates early origins is weaker: SNS-seq is a
 controversial method and even accepting the data shown in Fig. 4 at face value, only a few of the very
 strongest origins are affected. Nor is any data presented indicating that Rif1 interacts specifically or

directly with origin sites. Still, the interpretation of this SNS-seq experiment does not affect the main
 conclusions of the paper concerning Rif1 association domains and effects on timing. The authors go
 on to show that MCM and Rif1 act additively in B cells to affect early RT and early origin activity, with
 MCM having a noticeably stronger effect. Perhaps surprisingly, even depleting both MCM and Rif1
 does not affect chromatin organisation. Finally, the authors make the interesting observation that in
 Rif1^{-/-} or shMcm6 B cells the regions whose replication timing is most compromised are those that
 show the highest levels of transcription. Based on this last finding the authors provide an interesting
 Discussion suggesting reasons why Rif1 might have been re-deployed in B cells to have this special
 effect on replication, and relate their findings to an earlier study where human Rif1 was shown to
 stabilise Orc1 on chromatin to promote origin licensing. Overall, given the interesting new findings the
 paper is a good candidate for Nature Communications. The following points should be addressed:

We agree that RIF1 has a very weak effect on regulating origin activity. We have modified the text,
 abstract and discussion accordingly. We also mention in the revised Discussion that SNS-seq
 does not detect all origins, but probably only the stronger ones, meaning that there remains the
 possibility that RIF1 may have a broader role in origin activity.

Major Points

1. In places the textual explanations are insufficient and/or opaque for the non-expert. In several
 places more detailed explanation should be provided explaining the aspects on which the reader
 should focus (e.g., for Fig. 1C, Fig. 2E and Fig. 3E).

We apologize for this. We have explained these in more detail in the revised version (see blue text
 in pages 5, 7, 8 and 9).

2. The Title and Abstract should specify that the study examines mouse cells.

We have made changes to the title and abstract to specify the use of mouse cells.

3. Although it's clear that the major effect of Rif1 KO is on early regions, there is also some
 advancement on the replication time of late regions, seen for example in Fig. 1B and 1D. This effect,
 consistent with previous studies of Rif1, is overly downplayed in the main text.

We have explained this better in the revised version (pages 5-6), but in essence, we think that this
 is an indirect effect of the decreased efficiency of early origins in the KO cells.

4. Saddle plots in Fig. 2F-G seem to over-represent/exaggerate the changes in chromatin contacts
 that are seen in Fig. 2E. Therefore, Fig. 2F-G could be omitted.

We agree with the reviewer but given that such plots are widely used by the genome architecture
 community to characterize changes in compartmentalization, we anticipate that several readers
 would expect to see this, and therefore, we have retained it.

5. The co-localisation of Rif1 peak intensity with transcription, chromatin accessibility and the
 H3K27ac mark is mild/weak in each case in Fig. 3E. This should be mentioned in the text.

We have provided a more detailed analysis of the heatmap in Fig. 3E using k-means clustering,
 which better addresses the relationship of RIF1 peaks with transcription and chromatin features.

6. Fig. 4B uses SNS-seq to identify sites of initiation, which is controversial as the technique may
 assign some sites of re-priming as origins of replication. Given this issue, the mildness of the effect in

Fig. 4B, and the fact that Rif1 does not bind origin sites directly but rather early domains, the authors
should alter their abstract to remove the words 'regulates early origin firing', and moderate their claims
that Rif1 directly controls origin initiation.

We have amended the abstract as requested and we have made it explicit in the revised text that
RIF1 only regulates ~5% of ISs and hence that it is not a major regulator of origin activity. See also
see our response in lines 273-276 above.

7. The meaning of the vertical 'lines' observed through the centre of the DHS-seq and H3K36me3
experimental datasets in Fig. 4B is unclear. Could the authors elaborate on the reason for these low
signal zones?

These low-signal regions are most likely the nucleosome-free regions where the ISs are most
abundant. This agrees with previous reports showing that ISs occur in the space between two
nucleosomes, which is also where OCCM complexes would reside.

8. It would be helpful to move Fig. S4G into Fig. 5 to show the density-contour plots of Rif1^{-/-} and
Rif1^{-/-} shMcm6 cells together with their effects on RT and chromatin architecture (using Hi-C).

We have done as requested.

9. The scatter plot in Fig. 6A regarding Rif1^{-/-} shMcm6 cells is not accompanied by the same
heatmaps as in Fig. 4B for Rif1^{-/-} cells. This data should be included in Fig. 6.

We have provided the heatmap as requested (new Fig, 6B).

10. Could the authors test whether the positive effect of Rif1 on early origin firing is dependent on
PP1? This would further strengthen their model as outlined in the discussion section.

Please see our explanation for reviewer 2, who also raised this point. We explain why testing the
requirement of PP1 is technically very challenging and would take well over a year, if successful.
More importantly, even if we were to find a PP1-dependency, we do not know of any candidate
RIF1-PP1 target of dephosphorylation in B cells, which is essential to address mechanistic
questions.

Minor Points

1. Often 'shMcm6' labelling on figures is mislabelled as 'sMcm6'. (e.g., Fig. 1B and Fig. 2C).

Thank you for pointing out, we have corrected this.

2. An experimental condition label (i.e., shMcm6 vs. shLacZ) on top of each panel would make it
clearer what each experiment is showing (e.g., Fig. 1C).

Please note that in all figures, the axes labels explicitly state the experimental condition.

3. Fig. S1A and S1C and Fig. S3A-B show the reproducibility of each Repli-seq, SNS-seq or CHIP-
seq experiment, but are not particularly informative for the general reader and could be less
prominent.

We agree that they may not be informative to general readers, but we know that there are many
who will want to see such analyses to judge the reproducibility for themselves. Indeed, many

papers have such analyses. Hence, we have retained it in the supplemental section where its
presence is not disruptive to the flow.

4. The authors could associate an 'Early' or 'Late' label with the positive or negative values in each
figure where RT is examined (e.g., Fig. 1B). At least for this Reviewer, the graphs could be read more
intuitively with the x-axis reversed so 'Early' is on the left and 'Late' on the right.

We generated these plots following the convention in the field (for example, all papers by our co-
authors and leading RT experts, D. Gilbert and S. Buonomo, use this layout for showing RT).

Hence, we have chosen to stick to this style.

5. The blue and red colours used in Fig. 1F and 1G are confusing as they don't mean the same thing
(i.e., one denotes RT and the other denotes an experimental condition e.g., shMcm6).

Thank you for raising this. We have changed the colors in Fig. 1F and 1H.

6. In Fig. 2C, given the fairly minor changes in RT that occur in Rif1^{-/-} conditions (i.e. Early regions
somewhat delayed, but still Early) we wouldn't expect much of a change in PC1 values. So, the
conclusion drawn is perhaps not as surprising as the authors suggest and is somewhat overstated.

RT and genome organization are uncoupled, as we and others have shown earlier. Also, RIF1 was
shown to have more significant effects on genome organization in other cells. Given this
background, it was not predictable that loss of RIF1 in B cells would have the minor effects we
saw. In other words, it was theoretically possible that RIF1 could have had a more major effect on
genome compartmentalization even if the RT effect was small.

7. The lane order between Fig. 3C and Fig. 3D is changed, which is confusing when trying to compare
RT and Rif1 localisation in the primary B and MEF cell lines.

We have changed the order in Fig. 3C.

8. The inclusion of the word 'decreasing' in the y-axis label of Fig. 3E is confusing. The authors should
omit this word but keep the grey bar indicating peak intensity.

We have changed this heatmap with a new analysis of the same data, as mentioned above.

9. A couple of figures are mislabelled in the main text (e.g., Fig. 2F on page 7 should read Fig. 2G).
Fig. 4A is not referred to in the main text.

We have corrected this, thank you for pointing it out.

10. The data in Fig. 4B is presented with the 'strongest' origins at the bottom, which is confusing,
please invert these pile-up plots.

We have flipped the heatmaps as requested.

11. The black colour used in Fig. 6D to denote overlaps should be changed to be more translucent as
it masks the data underneath it.

As the reviewer correctly points out, the black proportion of the figure represents the overlapping
part of the signal of both tracks whereas the remaining color part shows the excess of one or the

other analyzed condition. Therefore, there is no underlying masked data within the black color
section (it simply represents the shared amount of the compared tracks).

12. Fig. 7A can be omitted as Fig. 7B makes the point in a clearer and more intuitive manner.

We understand the point, but we feel that Fig. 7A is very important to show how RT of the bins
within a given class are affected and to what degree, because all bins are shown separately and
the color change reflects the changes across the bins. This is not possible in Fig. 7B where all bins
are clubbed together. We do agree with the reviewer that Fig. 7A comes across as the more
intuitive analysis, but we feel that both representations are necessary for a comprehensive
analysis.

13. The sentence in lines 385-388 is confusing and should be rewritten to clarify (do the authors mean
to say 'earlier' rather than 'later' in line 388?)

Yes, indeed, we meant to say "earlier". Thank you for noticing it.

REVIEWER COMMENTS

Reviewer #1 (Remarks to the Author):

The authors address all of this reviewer's comments/suggestions. The revised manuscript is well written and the findings will be important contribution to the field.

Reviewer #2 (Remarks to the Author):

Authors responded to most of the comments, but have decided not to conduct the experiments that were requested.

4-1 Mechanistic insight into how Rif1 may directly regulate early firing origins is lacking in this manuscript.

The authors state the reason for not conducting the Rif1-PP1 mutant experiments. The major excuse was the time required for construction of the mutant cells. I understand the effort and time required for generation of the mutants. I believe that the effort is worth it to understand the mechanism of Rif1 mediated regulation of early replication in B cells and that the effect of PP1 mutation on B cell RT regulation would provide crucial information. However, in this case, I acknowledge the authors' rebuttal and would consent with authors' decision.

4-1 Regarding Mcm phosphorylation.

There are ways to detect the phosphorylation mediated by Cdc7, which would correlate with PP1 associated with Rif1. The followings are literatures detecting the phosphorylation of Mcm4 and Mcm2.

PMID: 28560864. Figure 2 Mcm4 is shifted upward (slow migration) by Cdc7 mediated phosphorylation.

PMID: 28273463. Figures 2 and 3 Mcm4 is shifted upward by Cdc7-mediated phosphorylation.

PMID: 25412417. Figure 5 shows that Mcm2 is shifted downward (fast migration) by Cdc7-mediated phosphorylation.

PMID: 34020950. Figure 4 Phosphorylated MCM2 at Ser40 (pMCM2) is used as Cdc7 phosphorylation target.

Basically, same patterns are observed for mice MCM as well.

Authors need to examine this by looking at MCM2 or MCM4. They could detect the phosphorylation of TMBP as well. The shift can be enhanced by using phosgel, if necessary. However, the shift should be visible on a regular gel.

S40,41 antibody of MCM2, can also be used as phosphorylated Cdc7 marker (PMID: 34020950).

If there is no enhancement of the phosphorylation in Rif1 depleted B cells, Rif1 may not interact with PP1 in B cells, which should also be examined.

4-3

"This is a good suggestion, but unfortunately, it is technically unfeasible. We have tried multiple times with several antibodies, but Mcm ChIP-seq has never worked for us."

Rif1 was previously reported to be required for efficient origin licensing (PMID: 28077461). Since authors propose that Rif1 loss leads to reduced licensing, causing less efficient early origin firing, I feel that this need to be tested. If the ChIP seq is difficult with currently available antibodies, authors could examine licensing by conducting immunostaining of Mcm after prewash with detergent and examine the effect of Rif1 depletion in B cells.

In both experiments described above, authors need to compare the results with a cell line that is known to undergo late to early changes of replication timing upon loss of Rif1.

Reviewer #3 (Remarks to the Author):

The authors have responded to the points raised and certainly adjusted the text and conclusions to take account of the major criticism that the original manuscript text was misleading given the mildness of any direct effect on early origins.

The additional k-means analysis of clustered data in Figures 3E-F is helpful, but in the case of the MEF data shown it is rather confusing whether the RIF1 peaks shown are within/correspond to the (generally late-replicating) RADs present in MEFS, particularly in light of the sentence "In agreement with the fact with most 254 RIF1 peaks in MEFs are in 255 early-replicating regions (Fig. 3C-D)...". So does this plot show analysis specifically of peaks that are not associated with the major RIF1 binding zones in MEFS?

It was disappointing that they dismissed many of the other suggestions that were aimed at clarifying the manuscript to make it more easily understood for the non-expert.

Response to Reviewer Comments

We thank the reviewers for their time and feedback. We have performed two new experiments to address Mcm4 phosphorylation in Rif1-KO B cells (new Fig. S1D) and whether RIF1 has a role in origin licensing in B cells (new Fig. S4B-C). We believe these new results have strengthened the paper by providing additional insights into the role of RIF1 in the B cell RT program. Please find below our point-by-point response in blue text. All new text in the main manuscript is in red.

Reviewer #1 (Remarks to the Author):

The authors address all of this reviewer's comments/suggestions. The revised manuscript is well written and the findings will be important contribution to the field.

Thank you for appreciating our work.

Reviewer #2 (Remarks to the Author):

Authors responded to most of the comments, but have decided not to conduct the experiments that were requested.

4-1 Mechanistic insight into how Rif1 may directly regulate early firing origins is lacking in this manuscript.

The authors state the reason for not conducting the Rif1-PP1 mutant experiments. The major excuse was the time required for construction of the mutant cells. I understand the effort and time required for generation of the mutants. I believe that the effort is worth it to understand the mechanism of Rif1 mediated regulation of early replication in B cells and that the effect of PP1 mutation on B cell RT regulation would provide crucial information. However, in this case, I acknowledge the authors' rebuttal and would consent with authors' decision.

Thank you for understanding this.

4-1 Regarding Mcm phosphorylation.

There are ways to detect the phosphorylation mediated by Cdc7, which would correlate with PP1 associated with Rif1. The followings are literatures detecting the phosphorylation of Mcm4 and Mcm2.

PMID: 28560864. Figure 2 Mcm4 is shifted upward (slow migration) by Cdc7 mediated phosphorylation. (MCM10A cells)

PMID: 28273463. Figures 2 and 3 Mcm4 is shifted upward by Cdc7-mediated phosphorylation. (Xenopus and HeLa)

PMID: 25412417. Figure 5 shows that Mcm2 is shifted downward (fast migration) by Cdc7-mediated phosphorylation. (both human cancer lines).

PMID: 34020950. Figure 4 Phosphorylated MCM2 at Ser40 (pMCM2) is used as Cdc7 phosphorylation target. (looks like all cancer cell lines).

Basically, same patterns are observed for mice MCM as well.

Authors need to examine this by looking at MCM2 or MCM4. They could detect the phosphorylation of TMBP as well. The shift can be enhanced by using phosgel, if necessary. However, the shift should be visible on a regular gel.

S40,41 antibody of MCM2, can also be used as phosphorylated Cdc7 marker (PMID: 34020950).

If there is no enhancement of the phosphorylation in Rif1 depleted B cells, Rif1 may not interact with
PP1 in B cells, which should also be examined.

As requested, we have probed for MCM4 phosphorylation in the chromatin fraction as done in
previous studies, and we found that there is indeed a slower-migrating population in Rif1-KO cells
but not in WT (new Fig. S1D). Moreover, this shift is seen in WT cells when the PP1 inhibitor
Tautomycin is added. Finally, the RIF-PP1 interaction has been recently reported in murine B cells
(PMID: 35416772). These observations suggest that the absence of the late replication phenotype
in B cells is not associated with deficient MCM4 dephosphorylation via RIF1-PP1. Thank you for
suggesting this experiment.

4-3

"This is a good suggestion, but unfortunately, it is technically unfeasible. We have tried multiple times
with several antibodies, but Mcm ChIP-seq has never worked for us."

Rif1 was previously reported to be required for efficient origin licensing (PMID: 28077461). Since
authors propose that Rif1 loss leads to reduced licensing, causing less efficient early origin firing, I
feel that this need to be tested. If the ChIP seq is difficult with currently available antibodies, authors
could examine licensing by conducting immunostaining of Mcm after prewash with detergent and
examine the effect of Rif1 depletion in B cells.

In both experiments described above, authors need to compare the results with a cell line that is
known to undergo late to early changes of replication timing upon loss of Rif1.

To address this question, we have performed ChIP-qPCR for MCM5 using an antibody which we
have previously shown to work for ChIP-qPCR (but not ChIP-seq) (PMID: 36108018). Since only
5% of ISs show >1.5-fold change in activity in Rif1-KO cells, we selected five of the most
downregulated ISs and compared them with four unchanged ISs. We found that the
downregulated ISs had decreased MCM5 occupancy but that the unchanged ISs did not show any
change in MCM5 association (new Fig. S4B-C). Therefore, although RIF1 does not regulate
replication origin activity at the vast majority of origins in B cells, a small subset of origins requires
RIF1 for optimal firing efficiency. Thank you for prompting this experiment. It is a useful addition to
the study.

In the experiment discussed in response to the previous comment (lines 44-50; new Fig. S1D), we
find that the bulk levels of chromatin-associated MCM4 are not changed in Rif1-KO B cells,
arguing against a global licensing defect. This result is consistent with the SNS-seq data where
only 5% of origins show >1.5-fold changes in activity. Please note that although the study
mentioned by the reviewer (PMID: 28077461) did implicate RIF1 in licensing in 293 cells, there are
other reports, in addition to our present one, that did not observe such a relationship. For example,
studies from our co-author have shown that MCM3 levels were not changed on chromatin by
western blot in Rif1-KO MEFs (PMID: 22850673) or by immunofluorescence in Rif1-KO ES cells
(PMID: 34006872). Studies in RIF1-depleted HeLa cells (PMID: 28273463 and 22850674) also did
not report changes in MCM or ORC in the chromatin fraction by western blot. Thus, the effect of
RIF1 on global licensing is not generalizable across different systems and may be cell type-
dependent.

**Reviewer #3 (Remarks to the Author):**

The authors have responded to the points raised and certainly adjusted the text and conclusions to
take account of the major criticism that the original manuscript text was misleading given the mildness
of any direct effect on early origins.

The additional k-means analysis of clustered data in Figures 3E-F is helpful, but in the case of the
MEF data shown it is rather confusing whether the RIF1 peaks shown are within/respond to the
(generally late-replicating) RADs present in MEFs, particularly in light of the sentence "In agreement
with the fact with most RIF1 peaks in MEFs are in early-replicating regions (Fig. 3C-D)...". So does

this plot show analysis specifically of peaks that are not associated with the major RIF1 binding zones
in MEFS?

It was disappointing that they dismissed many of the other suggestions that were aimed at clarifying
the manuscript to make it more easily understood for the non-expert.

We apologize for the confusion, and we have clarified this in the revised manuscript. For the
analysis of peaks in Fig. 3C-F, we only define peaks as those that fall outside of RADs. If a peak
was called inside a RAD, it is considered as part of that RAD. In other words, peaks and RADs are
distinct, non-overlapping regions in our analyses. We have added this explanation to the revised
manuscript. In MEFs, the peaks are mostly early-replicating and RADs are mostly late-replicating.
Importantly, both peaks and RADs are bona fide RIF1 binding regions, but their distribution in the
genome is different between MEFs and B cells.

Regarding other suggestions, we believed we had addressed all the points made by the reviewer.
For example, we had provided more details for Fig.1C, Fig. 2E and Fig. 3E, as explicitly mentioned
by the reviewer (please see major point 1 from the reviewer in the first review). We went back and
checked all our responses, but we did not find anything that was not responded to.

The reviewer had asked us to remove some analyses, such as saddle plots and correlation
matrices, but we had also explained why these analyses were retained, namely, because they are
widely used and would be expected by many readers. We hope the reviewer understands these
points.

REVIEWERS' COMMENTS

Reviewer #2 (Remarks to the Author):

Authors responded to my comments by conducting additional experiments. In Figure S4, authors showed significant decrease in MCM5 enrichment at all the five downregulated ISs, but not at the unchanged ISs. However, this has not been discussed much in terms of effect of Rif1 on early RT.

Although only ~5% of ISs showed >1.5-fold change in read density upon loss of RIF, does this result suggest that Rif1 does affect licensing in activated B cells? And could that be a part of the reason for shift to later replication of early RT in Rif1 KO cells? The cumulative effects of Rif1KO on licensing over the entire genome, whilst the effect at each origin may be small, could affect early replication, since reduced (or delayed) licensing in G1 phase could lead to late firing.

I wish authors conducted MCM5 ChIP-seq and examine the correlation between MCM5 binding and Rif1 binding (as well as RT), which would provide more information on potential effect of Rif1 on licensing efficiency and its relation to RT change. (I understand from the letter that the current MCM5 antibody does not work for ChIP seq.)

At least authors could discuss the above possibility in the text.

Figure 5D legend

A magnified view of the boxed region in C.

->

There is no box in Figure 5C.

REVIEWERS' COMMENTS

Reviewer #2 (Remarks to the Author):

Authors responded to my comments by conducting additional experiments. In Figure S4, authors showed significant decrease in MCM5 enrichment at all the five downregulated ISs, but not at the unchanged ISs. However, this has not been discussed much in terms of effect of Rif1 on early RT.

Although only ~5% of ISs showed >1.5-fold change in read density upon loss of RIF, does this result suggest that Rif1 does affect licensing in activated B cells? And could that be a part of the reason for shift to later replication of early RT in Rif1 KO cells? The cumulative effects of Rif1KO on licensing over the entire genome, whilst the effect at each origin may be small, could affect early replication, since reduced (or delayed) licensing in G1 phase could lead to late firing.

I wish authors conducted MCM5 ChIP-seq and examine the correlation between MCM5 binding and Rif1 binding (as well as RT), which would provide more information on potential effect of Rif1 on licensing efficiency and its relation to RT change. (I understand from the letter that the current MCM5 antibody does not work for ChIP seq.)

At least authors could discuss the above possibility in the text.

Please note that in the Discussion, we had mentioned the possibility that RIF1 may play a broader role in early origin licensing given that SNS-seq may not identify weak or infrequently used origins. However, we based our main conclusion on the ISs that we can detect (Fig. 4a-b). Here, it is apparent that the vast majority of origins do not show major changes in activity. For this reason, we concluded that RIF1 is not a major contributor to origin licensing in B cells. We have now added an additional line in the Discussion acknowledging the possibility that if RIF1 were to regulate more early origins that those detected by SNS-seq, then the early replication phenotype could be, in part, the result of decreased RIF1-mediated early origin licensing. Thank you for suggesting this.

Figure 5D legend

A magnified view of the boxed region in C.

-> There is no box in Figure 5C.

We have corrected this. Thank you for pointing it out.